# Cell2fate infers RNA velocity modules to improve cell fate prediction

Alexander Aivazidis[1], Fani Memi [1], Vitalii Kleshchevnikov [1], Sezgin Er [2], Brian Clarke [3], Oliver Stegle [1,3,4] ✉ & Omer Ali Bayraktar [1] ✉

RNA velocity exploits the temporal information contained in spliced and unspliced RNA counts to infer transcriptional dynamics. Existing velocity models often rely on coarse biophysical simplifications or numerical approximations to solve the underlying ordinary differential equations (ODEs), which can compromise accuracy in challenging settings, such as complex or weak transcription rate changes across cellular trajectories. Here we present cell2fate, a formulation of RNA velocity based on a linearization of the velocity ODE, which allows solving a biophysically more accurate model in a fully Bayesian fashion. As a result, cell2fate decomposes the RNA velocity solutions into modules, providing a biophysical connection between RNA velocity and statistical dimensionality reduction. We comprehensively benchmark cell2fate in real-world settings, demonstrating enhanced interpretability and power to reconstruct complex dynamics and weak dynamical signals in rare and mature cell types. Finally, we apply cell2fate to the developing human brain, where we spatially map RNA velocity modules onto the tissue architecture, connecting the spatial organization of tissues with temporal dynamics of transcription.

The concept of 'RNA velocity', which involves inferring transcriptional dynamics from spliced and unspliced counts in single-cell RNA sequencing (scRNA-seq), has displayed notable potential[1–4]. The first implementations of RNA velocity models[1–3] have undergone an evolution of conceptual and technical refinements, including improved parameter inference[5–7] as well as the use of numerical approaches[6,8–10] to solve the underlying differential equations. However, these existing refinements are bound to tradeoffs between either introducing coarse biophysical approximations[1–3,5,7,11,12] or relying on extensive numerical approximations[6,8–10]. Hence, the fundamental challenge remains to define a mathematically sound framework that allows for capturing realistic transcriptional dynamics while retaining computational and numerical tractability.

To address the aforementioned limitations, we present cell2fate, a fully Bayesian model of RNA velocity based on a more realistic biophysical model of complex transcription dynamics. Cell2fate employs linearization to decompose differential equations describing complex transcriptional patterns, such as transcriptional boosts, into tractable components that can be solved analytically. By doing so, the model is at the same time expressive, interpretable and computationally efficient. The approach to decompose the velocity problem into components also provides a connection between RNA velocity and dimensionality reduction using a biophysical solution.

We assess and benchmark cell2fate in the context of different real-world settings, demonstrating its ability to capture complex dynamics and weak dynamical signals in rare and mature cell types. Finally, we show how cell2fate can be combined with spatial transcriptomics, thereby connecting transcriptional dynamics to their spatial tissue environment.

## Results

### The cell2fate model
Cell2fate builds on established concepts for RNA velocity[1,2], employing a dynamical model to explain variation in spliced ($s$) and unspliced ($u$)

[1]Wellcome Sanger Institute, Cambridge, UK. [2]International School of Medicine, Istanbul Medipol University, Istanbul, Turkey. [3]Division of Computational Genomics and Systems Genetics, German Cancer Research Center (DKFZ), Heidelberg, Germany. [4]Genome Biology Unit, European Molecular Biology Laboratory, Heidelberg, Germany. ✉e-mail: oliver.stegle@embl.de; ob5@sanger.ac.uk

read counts for individual genes and cells (Fig. 1a), which can be defined in two coupled ODEs:

$$\frac{du_g}{dt} = \alpha_g(t) - \beta_g u_g \tag{1}$$

$$\frac{ds_g}{dt} = \beta_g u_g - \gamma_g s_g. \tag{2}$$

Here, $\alpha$, $\beta$, $\gamma$ denote the transcription, splicing and degradation rates for different genes $g$. Solving the ODEs for $u$ and $s$ and fitting the equations to observed counts allows estimation of the unknown rate parameters, which can then be substituted into equation (2) to obtain the rate of change in spliced counts in each cell, $\frac{ds}{dt}$, which in turn gives rise to what is commonly referred to as 'RNA velocity' (ref. 1). An important challenge for RNA velocity models is that transcription rates as a function of differentiation time ($\alpha_g(t)$) can be complex and nonlinear, reflecting the implicit dependency on active transcription factor (TF) abundance in the nucleus[13], yet the integral of the transcription rate function $\alpha_g(t)$ needs to remain tractable to allow the ODEs to be solved efficiently. As a consequence, existing methods either assume simplified stepwise functions for $\alpha_g(t)$ (refs. 1,2,5,7) or they resort to numerical approximations to solve the ODE[6,8–10] (Supplementary Notes and Supplementary Fig. 3).

In cell2fate, we use an expansion of the derivative of the transcription rate in terms of individually integrable basis functions, which we refer to as linearization in the following:

$$\frac{d\alpha_g}{dt} = \sum_{m=1}^{M} \frac{d\alpha_{mg}}{dt} = \sum_{m=1}^{M} \lambda_{mi}(\hat{\alpha}_{mgi} - \alpha_{mg}). \tag{3}$$

We term each basis function a module, denoted by a subscript $m$. Subscript $i$ denotes the state of a module (ON or OFF). $\hat{\alpha}_{mgi}$ is the target transcription rate of a module, which takes on nonzero values for all genes, when the module is in the ON state (Fig. 1b, bottom right). $\lambda_{mi}$ is the rate at which the target transcription rate is reached, and the state $i$ depends on switch times $T_{m,\text{ON}}/T_{m,\text{OFF}}$ on a cell-specific timescale $T_c$ (Fig. 1b, top). This choice of basis functions allows for each individual ODE as well as their total sum to be solved analytically (Methods and Supplementary Notes). The parameters $T_{m,\text{ON}}/T_{m,\text{OFF}}$, $\lambda_{mi}$ and $T_c$ are shared across all genes, which vastly reduces the number of parameters that need to be estimated compared to existing models yet still provides gene-specific transcription rates $\alpha_{mg}$.

In addition to being appealing for computational reasons, the linearization also provides a biophysical connection between RNA velocity and statistical dimensionality reduction. This link becomes apparent when casting the linearization as a mixed membership model, in which transcription rates, RNA velocity and spliced and unspliced counts of each gene are governed by a linear combination of $M$ modules (Methods and Fig. 1b). The mixing coefficients can then be interpreted analogously to gene loadings of factor analysis or principal-component analysis. Mechanistically, modules can be interpreted as approximating the transcription rate changes induced by all active regulatory proteins as a small set of independent effects that are each valid within time windows defined by $T_{m,\text{ON}}/T_{m,\text{OFF}}$.

Cell2fate is a fully Bayesian model that is fit to raw cell-level counts as input, and it includes a series of refinements to account for technical sources of variation, including overdispersion, variation in detection sensitivity of spliced and unspliced RNA molecules, ambient RNA and known batches (Methods). A hierarchical prior structure achieves efficient regularization of model parameters while sharing evidence strength across genes (Methods). The model is implemented in the probabilistic programming language Pyro[14] and fit using Pyro's stochastic variational inference framework using a customized variational distribution structure to account for dependencies between model parameters (Methods). The software implementation comes with guidelines and heuristics to determine hyperparameters such as the number of modules (Methods), and it builds on scvi-tools[15] to facilitate its integration in existing workflows.

## Improved predictions of cell fates and transcription rates

To benchmark the performance of cell2fate, we compared the model to existing RNA velocity methods by assessing the consistency of estimated cell fate trajectories with prior knowledge. Briefly, we considered the cross-boundary directional correctness (CBDir) metric to benchmark alternative models, thereby scoring the consistency of transition probabilities at the boundary between cell clusters with known transitions[3] (Methods).

We considered ten RNA velocity methods spanning different model classes and approaches for parameter inference (Table 1 and Methods). All methods were applied to five scRNA-seq datasets, including widely used benchmark datasets such as the developing mouse dentate gyrus[16] and pancreas[17] (Supplementary Fig. 1). To test the ability of different methods to resolve complex transcriptional dynamics, we also examined mouse erythroid maturation[18] and human bone marrow[19]: two datasets that feature multiple transcriptional boosts across cellular trajectories[12]. Finally, we considered a mouse bone marrow dataset with markedly low unique molecular identifier (UMI) counts[1], thereby assessing the extent to which these models cope with low-coverage data.

On average, across all five datasets, cell2fate achieved the best CBDir scores. More importantly, cell2fate inferred the correct directionality of cell fate transitions in all datasets, whereas all other methods, with the notable exception of pyroVelocity_model2, inferred reverse-order dynamics in at least one benchmark setting (corresponding to negative CBDir values). Inspecting the benchmarking results, we could attribute the performance of cell2fate to overcoming two major challenges as elaborated below.

First, cell2fate provided sufficient statistical power to identify correct velocity flows from subtle transcriptional dynamics. For example, in the mouse dentate gyrus dataset, other methods failed to resolve the late maturation trajectory of granule neurons, thus incorrectly inferring that mature cells transitioned into their immature counterparts (Fig. 2b, blue inset boxes and Supplementary Fig. 2). The same result was observed when applying CellRank[20] to the velocity estimates instead of uniform manifold approximation and projection (UMAP), indicating that only cell2fate could resolve the trajectory toward mature granule neurons (Supplementary Figs. 3 and 4).

Second, cell2fate correctly reconstructs complex transcriptional dynamics. In the mouse erythroid maturation datasets, the model resolved the correct cell trajectories, whereas other models tended to perform poorly (Fig. 2c and Supplementary Fig. 5). Previous analysis[18], based on the visual inspection of spliced and unspliced counts across manually annotated cell clusters, has provided evidence that mouse erythroid lineage formation features many 'multi-rate kinetic' genes such as *Hba-x* and *Nudt4* that display coordinated changes in transcription rates across the cell maturation trajectory[18]. Consistently, cell2fate recapitulated the stepwise transcriptional rate boosts in these multi-rate kinetic genes[18] (Fig. 2d, turquoise line). By contrast, other methods, such as pyroVelocity_model2, can only predict a single nonzero transcription rate, due to their simpler underlying dynamical model (Fig. 2d, green line), results that we again confirmed using CellRank (Supplementary Figs. 6 and 7).

As an alternative metric to CBDir, we also assessed concordance of the estimated time differences between clusters with prior knowledge (Supplementary Table 8), and we compared time outputs to known developmental ages of different mouse samples (Supplementary Figs. 8 and 9); these alternative metrics and benchmarks confirmed the performance benefits of cell2fate. Notably, granule neuron subclusters

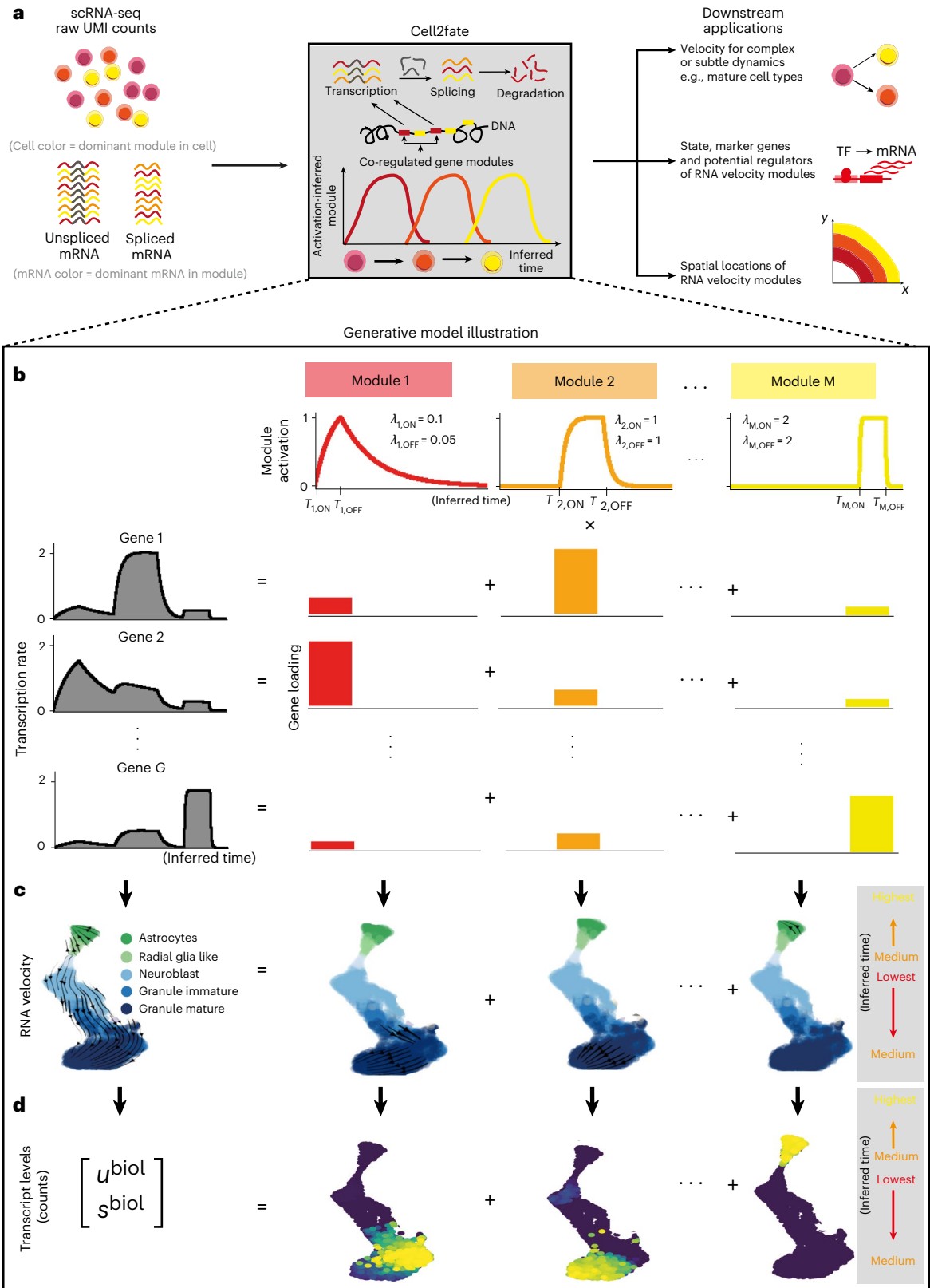

**Fig. 1 | Cell2fate model overview. a–d,** Cell2fate allows inferring complex and subtle transcriptional dynamics (**a**) by modeling gene-specific transcription rates (**b**) using a smaller number of independent modules with simple dynamics that also give rise to a modular structure in RNA velocity (**c**) and counts (**d**). $\lambda$ denotes the rate of module activation (ON) or deactivation (OFF). The superscript 'biol' denotes that counts ($u$ and $s$) do not include technical factors of variation, such as sequencing depth.

**Table 1 | Summary of RNA velocity model versions and parameter settings used in benchmark**

| Model | Version | Parameter settings |
|---|---|---|
| scVelo_dynamical | 0.2.5 | n_jobs=8 |
| scVelo_stochastic | 0.2.5 | n_jobs=8 |
| pyroVelocity_model1 | 0.1.0 | batch_size=4,000, use_gpu=0, cell_state='state_info', include_prior=True, offset=False, library_size=True, patient_improve=1×10⁻³, model_type='auto', guide_type='auto_t0_constraint', train_size=1.0 |
| veloVAE | N/A | tmax=20, dim_z=5, device='cuda:0' |
| UniTVelo_independent | 0.2.5 | R2_ADJUST=True, IROOT=None FIT_OPTION='2', GPU=0, vgenes = 'offset' |
| UniTVelo_unified | 0.2.5 | R2_ADJUST=True, IROOT=None FIT_OPTION='1', GPU=0, vgenes='offset' |
| DeepVelo | 0.2.5-rc.1 | deepVelo.Constants.default_configs |
| pyroVelocity_model2 | 0.1.0 | batch_size=4,000, use_gpu=0, cell_state='state_info', include_prior=True, offset=False, library_size=True, patient_improve=1×10⁻³, model_type='auto', guide_type='auto_t0_constraint', train_size=1.0 |
| VeloVI | 0.1.1 | N/A |

N/A in the 'Version' column means that a package did not have a version number associated with it on github. N/A in the 'Parameter settings' column means that a method did not require any additional parameter inputs other than the default settings.

with higher cell2fate estimated time contained more cells from the later developmental ages (Supplementary Fig. 8b), validating cell2fate estimates. Alternative methods as well as standard diffusion pseudotime analysis did not resolve this granule neuron maturation trajectory (Supplementary Figs. 8c and 10). Furthermore, cell2fate was robust to changes in the assumptions on prior distributions (Supplementary Fig. 11), and we confirmed the ability of the model to estimate ground truth dynamical rates when applying the model to semisynthetic data (Supplementary Fig. 12). Finally, we note that, while cell2fate has higher computational requirements than some existing methods, the computational requirements of the model are aligned with alternatives so that cell2fate can be readily applied to larger datasets (Supplementary Tables 6 and 7 and Supplementary Fig. 13).

We note that cell2fate's cell-specific timescale[3,6,7] (Fig. 1a,b) provides two additional use cases that can help to gain deeper insights. First, it aids the identification of cell lineage progression and distinct cell lineages. For example, in the mouse dentate gyrus dataset, granule neurons and astrocytes were assigned markedly disconnected time points, with oligodendrocytes occupying a mid-time point range (Fig. 2e, left), consistent with the distinct lineage origins of these three cell types[16]. By contrast, in the mouse erythroid maturation dataset, a single lineage with a single connected time range is identified (Fig. 2e, right). Second, inspecting the Bayesian posterior[5–7] of the cell-specific time provides a principled measure of confidence in the RNA velocity values across and within datasets. In both datasets mentioned above, the coefficient of variation (CV) of the posterior distribution of individual cell times was consistently estimated to be close to zero, indicating low uncertainty (Fig. 2e, bottom). By contrast, cell2fate applied to a steady-state dataset of peripheral blood mononuclear cells[21], in which no consistent transcriptional dynamics are expected[12], results in confidence estimates with a CV close to 1, indicating high uncertainty (Fig. 2f). The posterior uncertainties can also be used to estimate confidence levels in individual transitions. To do so, cell2fate implements a heuristic confidence score based on the fraction of posterior samples from the cell-specific time in one cluster that are higher than the 90th percentile of samples from another cluster. This heuristic

correlates well with the CBDir score of the corresponding transitions ($\rho = 0.56$), indicating that it is well calibrated (Supplementary Fig. 14). Hence, the posterior of cell-specific time can serve as quality control to assess whether cell2fate identifies meaningful dynamics in a given dataset.

In sum, our benchmark demonstrates cell2fate's enhanced statistical power to estimate cell trajectories and resolve complex transcriptional dynamics and the ability to quantify uncertainty in velocity estimates.

## RNA velocity modules map fine stages of late cell maturation

Cell2fate modules are sequentially activated gene expression programs over time. Given their biophysical foundations in transcriptional kinetics, we expect that RNA velocity modules can provide a more granular characterization of dynamic processes during cellular differentiation than conventional dimensionality reduction techniques that lack a mechanistic basis, such as matrix factorization or clustering. In addition, cell2fate comes with a suite of downstream analysis and visualization tools, enabling users to explore dynamic processes and derive biological insights.

To demonstrate the cell2fate toolkit, we considered the mouse brain single-cell dataset included as part of the benchmarking study (Fig. 2b), profiling the dentate gyrus region in the hippocampus across two developmental stages[16]. In addition to early differentiation of neurons and astrocytes from neural progenitors, this dataset covers the late maturation trajectory of granule neurons (that is, the late differentiation after the immature neuron stage), a critical process that is however not well understood, and, more generally, it is unknown whether this late maturation process unfolds across successive transcriptional stages. Previous RNA velocity methods applied to this dataset[2,3] were able to distinguish neuronal versus astrocyte lineage trajectories; however, the correct trajectory for the most mature granule neurons could not be resolved (Fig. 2b).

Cell2fate applied to this dataset revealed 16 distinct RNA velocity modules (Supplementary Fig. 15), capturing all the expected cell trajectories, with the dominant lineage corresponding to granule neuron differentiation and maturation stemming from neural intermediate progenitor cells (nIPCs), neuroblasts and immature neurons. Radial glial-like progenitor cells are largely committed to astrocytes (Fig. 3a), as evident both from cell2fate's time estimates and CellRank fate probabilities (Supplementary Fig. 16). We also observed that mossy cells, another neuronal population in the dentate gyrus, were assigned to the middle stages of the granule neuron trajectory. While mossy cells are thought to have different lineage origins, their transcriptional development is highly similar to that of granule neurons[22].

To explore the dynamics of neuronal differentiation in greater depth, we used the fitted cell2fate model to estimate the total spliced transcript abundance for each of the nine granule neuron lineage modules in individual cells across the inferred time (Fig. 3d, top). This analysis identified the successive induction of modules across the early differentiation of radial glia into nIPCs, neuroblasts and immature neurons (modules 1–3). Strikingly, cell2fate also recovered dynamics in mature granule neurons, explained by six modules (modules 4–9) that are sequentially activated and temporally overlap across mature granule neurons, thereby finely dissecting the late maturation of these cells into distinct transcriptional windows (Fig. 3d). The model also correctly identified a temporal gap between immature and mature granule neurons (Fig. 3d), which is consistent with prior expectations[16]. The cell2fate visualization tool complements *t*-distributed stochastic neighbor embedding or UMAP by providing dynamic insights anchored on estimated differentiation time, and it can also visualize additional metadata such as cell type annotations or developmental age (Fig. 3d, bottom two panels).

The total spliced count estimates can also be used to visualize the dynamics of RNA velocity modules across cells, for example, on a

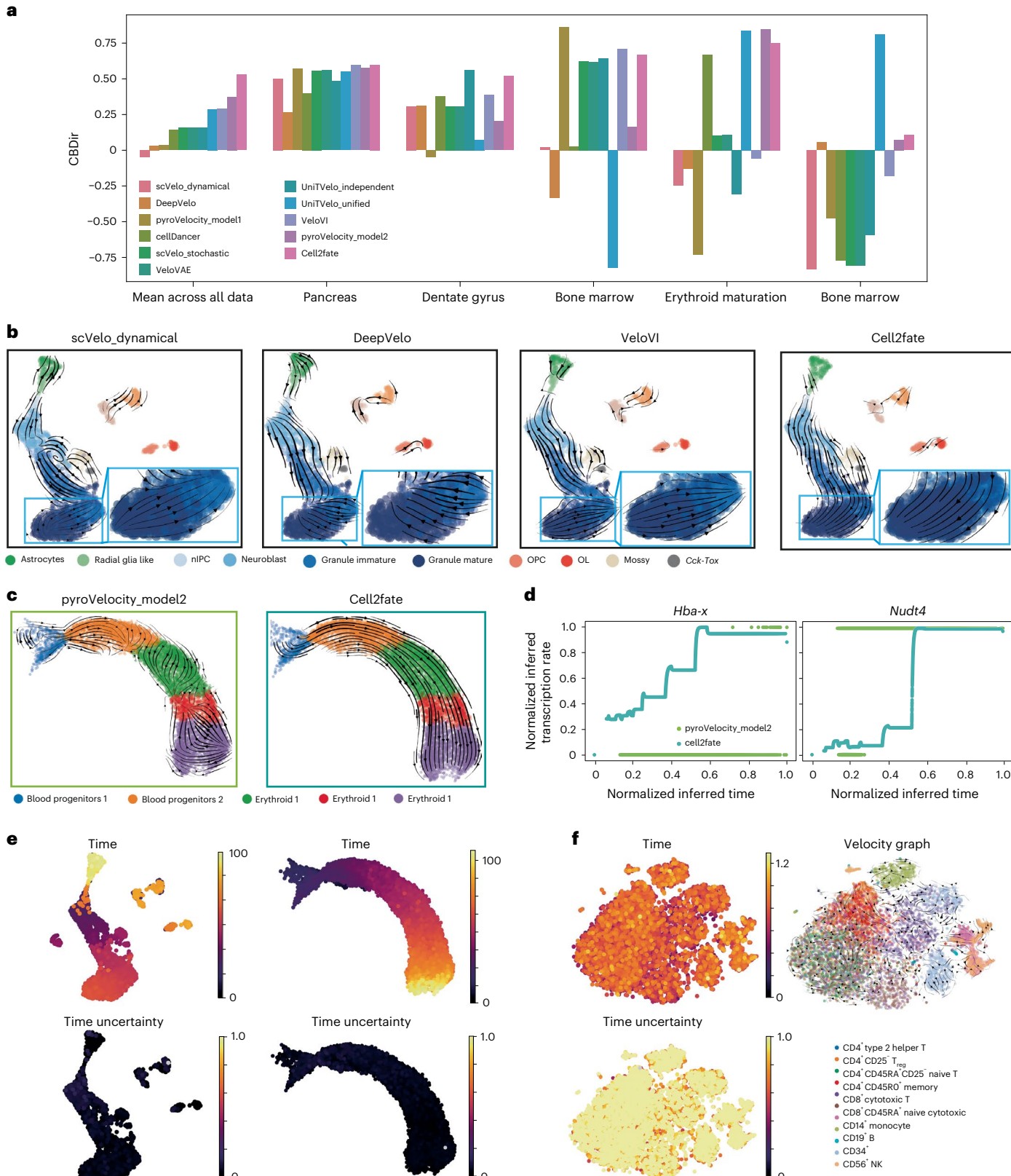

**Fig. 2 | Enhanced performance of cell2fate in benchmark of ten methods across five datasets. a**, Performance of ten methods to reconstruct known trajectories on five datasets. Shown is the CBDir[3], with positive values corresponding to correct lineage reconstructions. **b–d**, Examples for datasets that harbor specific challenges, including weak signals in mature cell types (OPC, oligodendrocyte progenitor cell; OL, oligodendrocyte) (**b**) and complex transcription rate dynamics (**c,d**). **d**, Transcription rate as inferred

for two selected genes that have been postulated to have stepwise changes in transcription rates[18]. **e**, Cell-specific time estimates from cell2fate. Left, astrocytes have much higher time than the neuron lineage. Right, one connected time range indicates a single lineage. **f**, CV of the posterior time can be used as an uncertainty measure to assess the suitability of a dataset for cell2fate analysis. In a steady-state dataset of peripheral blood mononuclear cells, this CV is close to 1 throughout. NK, natural killer; $T_{reg}$, regulatory T cell.

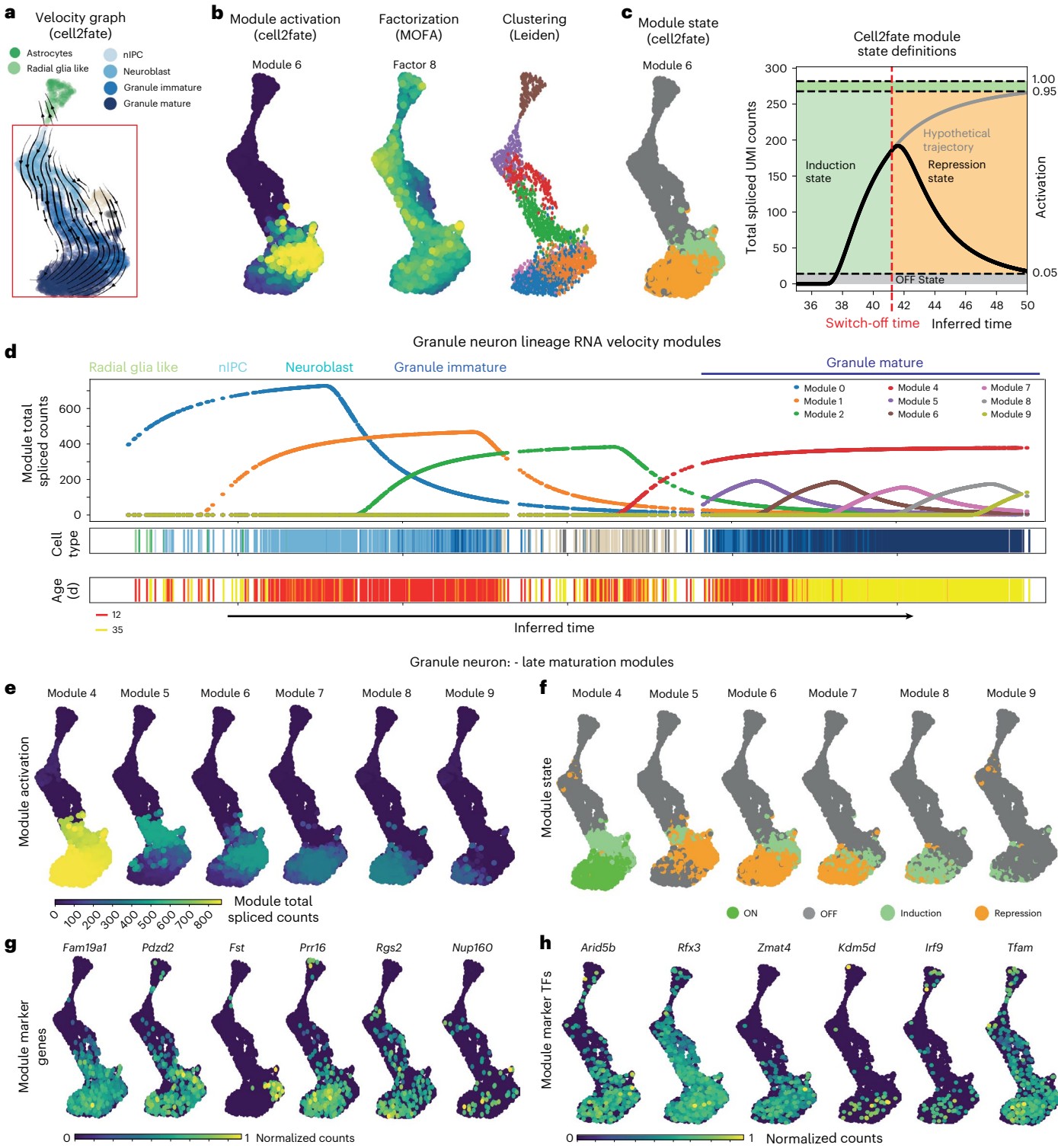

**Fig. 3 | Module decomposition of cell2fate resolves final stages of granule neuron maturation. a**, Cell2fate velocity graph embedding for dentate gyrus data. **b**, Spliced count abundance caused by selected modules over time in the granule neuron lineage. **c**, Activation of the selected example cell2fate module, weight of the selected MOFA[23] factor and Leiden clustering for comparison. **d**, Modules in each cell can be classified into states for easier interpretation and visualization, based on how much of their maximal steady-state counts they have reached and whether they are increasing or decreasing in expression. **e**, Activation, defined as the spliced counts produced by a module in each cell. **f**, State of late granule neuron maturation modules. **g**, Module marker genes, defined as having a large part of their transcription rate explained by this module. **h**, Module TF genes, defined as module marker genes that are also known TFs.

conventional UMAP plot (Fig. 3b). The activation of different modules per cell can be inspected, similar to the factor activity in conventional matrix factorization. We also compared these module activation estimates to conventional factor analysis and clustering methods. Briefly,

multi-omics factor analysis (MOFA)[23] yielded factors that captured complementary sources of variation, with activity profiles that were temporally more diffuse across the differentiation trajectory. Specifically, these factors did not stratify late granule neuron maturation

(Fig. 3b and Supplementary Fig. 17). We also observed overall low correlation between cell2fate module and MOFA factor gene loadings, particularly for the late neuronal maturation modules 4–9 (Supplementary Fig. 18). Similarly, Leiden clustering of the scRNA-seq dataset at different resolutions identified clusters that were not aligned with neuronal maturation (Fig. 3b and Supplementary Fig. 19). Collectively, these observations indicate that cell2fate captures complementary aspects of variation compared to existing decomposition methods and is well suited to conduct granular dissection of granule neuron maturation.

Beyond the activity of modules, the dynamics can be further classified into states within each cell, based on whether they are increasing or decreasing in expression (Fig. 3c and Methods). Both quantities are shown in Fig. 3e,f for granule neuron differentiation. The additional dynamic information in this visualization shows, for example, that module 9 has not reached a steady state, implying that granule neuron maturation continues beyond the time range captured in this dataset.

Finally, we examined to what extent RNA velocity modules can provide deeper insights into the late stages of granule neuron differentiation. We ranked genes by how much of their transcription rate is explained by each module and used top genes as 'module markers' (Fig. 3g and Supplementary Fig. 20). We identified *Rbm24*, Tafa1 (*Fam19a1*) and *Sptb* (module 4) and Pakap (*Palm2*), *Pdzd2* and *Usp19* (module 5) as markers switched on in immature granule neurons (Fig. 3g and Supplementary Fig. 20). By contrast, *Fst*, *Rapgef5* and *Moxd1* (module 6), *Prr16*, *Kdm5d* and *Rpa3* (module 7) and *Rgs2*, *Nudt13* and *1700048O20Rik* (module 8) provide markers of late granule neuron maturation stages (Fig. 3g). *Fam19a1* has been reported to suppress neural stem cell maintenance and promote differentiation[24,25], consistent with its expression pattern in maturing granule neurons. *Rgs2* is dynamically expressed during neuronal activity[26] and involved in synaptic plasticity[27], consistent with its late induction in module 8. Apart from these two genes, the marker genes reported here have not been functionally investigated in granule neurons or brain development to our knowledge. We further intersected the top 30 markers of each module with a set of 30 Alzheimer's risk genes[28] and 250 autism spectrum disorder risk genes[29,30], which highlighted late granule neuron maturation module 8, where the Alzheimer's associated gene *Trappc6a* and the autism-associated genes *Kdm6a* and *Nr4a2* were among the markers (Supplementary Table 9). These results highlight that cell2fate can lead to new biological insights, even in a widely characterized cell lineage like granule neurons.

Additionally, we can extract top module marker genes that are TFs as 'module TFs' (Fig. 3h). Moving on to such module TFs, *Zmat4* (module 6) and *Tfam* (module 8) are enriched in late granule neuron maturation stages (Fig. 3h). *Zmat4* has been reported as upregulated in the auditory cortex of young mice at postnatal day 7 compared to adults[31], while *Tfam* knockout results in immature neuronal phenotypes[32]. Yet their roles in granule neuron differentiation have not been studied to date. We also find that genes with putative promoter sequences that are most likely to be bound by the top 20 module TFs, as predicted by the ProBound algorithm[33], are more frequently among the top 300 module genes than those least likely to be bound by those TFs (Supplementary Fig. 21). These TFs provide putative candidate regulators of late granule neuron differentiation.

In sum, our results demonstrate the great interpretability and statistical power of cell2fate's module decomposition for scRNA-seq datasets to finely dissect cellular processes and suggest that late granule neuron maturation is composed of distinct stages.

## Spatial mapping of RNA velocity modules

Temporal biological processes are often spatially organized in tissues. For example, cell differentiation and migration are often coupled, with cells associating with distinct spatial signaling microenvironments throughout their differentiation trajectories. Here, we sought to link the temporal information captured by cell2fate to spatial tissue organization by mapping RNA velocity modules in a newly generated spatial transcriptomics dataset of human brain development (Fig. 4a).

We focused on the fetal human cerebral cortex in which excitatory neuron maturation follows a highly stereotyped trajectory through space and time[34]. Neural progenitors termed radial glia and intermediate progenitors reside in cortical germinal zones, where they sequentially give rise to distinct neuronal subtypes that subsequently migrate out to the deep and upper layers of the cortical plate across their maturation (Fig. 4b). Deep layer-residing neurons (DLn) are born before upper-layer neurons (ULn) in early gestation; hence DLn are relatively more mature than ULn by midgestation (Fig. 4b). Thus, the maturation state and spatial location of cortical excitatory neurons are tightly linked.

To examine cellular differentiation trajectories in the human cortex, we initially performed single-nucleus RNA-sequencing (snRNA-seq) profiling (10x version 3.0) of one donor at midgestation. We then followed standard snRNA-seq processing workflows (Methods) to cluster cells and annotated cell types using markers from the literature[35]. We annotated distinct neural progenitors (radial glial and intermediate progenitor cells) as well as excitatory neuron populations at different stages of maturation (Fig. 4c). As expected, mature neurons expressed DLn markers, whereas newborn and immature neurons showed enriched expression of ULn markers (Supplementary Fig. 22). We also annotated inhibitory neurons and glial cell types but excluded them from the subsequent excitatory neuron trajectory analysis.

We then applied cell2fate to this human brain snRNA-seq dataset and observed the expected excitatory neuronal differentiation trajectory from neural progenitors to newborn, immature and mature neurons (Fig. 4c). The RNA velocity modules dissected the neuronal trajectory into finer-grained maturation stages, identifying seven sequentially activated and temporally overlapping modules throughout immature and mature neurons (Fig. 4d and Supplementary Fig. 23). While these modules contained some DLn and ULn cell type markers, they also included many genes that are widely expressed across all excitatory neurons in the adult human cortex[36], such as *PSMC3*, *KRR1* and *BMPER* (Supplementary Figs. 24 and 25). This suggests that the modules partially identify a neuronal maturation trajectory common to both DLn and ULn.

In contrast to cell2fate, other RNA velocity methods such as scVelo were not able to accurately identify velocity flow in mature neurons (Supplementary Fig. 26). Additionally, the integrated measurement model of cell2fate allowed us to factor in different detection probabilities for spliced and unspliced counts and correct batch effects in our human brain snRNA-seq dataset (Supplementary Fig. 27), which is crucial for estimating true transcriptional dynamics from observed counts (Methods).

To spatially map our RNA velocity modules in the developing human cortex, we performed Visium spatial RNA-seq profiling (10x CytAssist) of one cortical tissue section from an age-matched donor (Fig. 4b). As the Visium assay offers coarse spatial resolution and profiles multiple cells at each tissue location (that is, Visium spot), we used the cell2location algorithm[37] to deconvolve the abundance of RNA velocity modules across spatial data. We used the steady-state expression counts of each module as reference gene expression signatures and then applied the standard cell2location workflow to infer the abundance of each module signature across Visium spots (Fig. 4a and the Methods).

The RNA velocity modules showed expected patterns of spatial mapping across the human cortex (Fig. 4e–f and Supplementary Fig. 28). Progenitor modules spatially mapped to germinal zones (Fig. 4e), while neuronal modules primarily mapped to the cortical plate (Fig. 4e and Supplementary Fig. 28). The fine spatial locations of neuronal modules were consistent with their maturation state (Fig. 4f). The immature ULn module (0) mapped to the upper cortical layers as

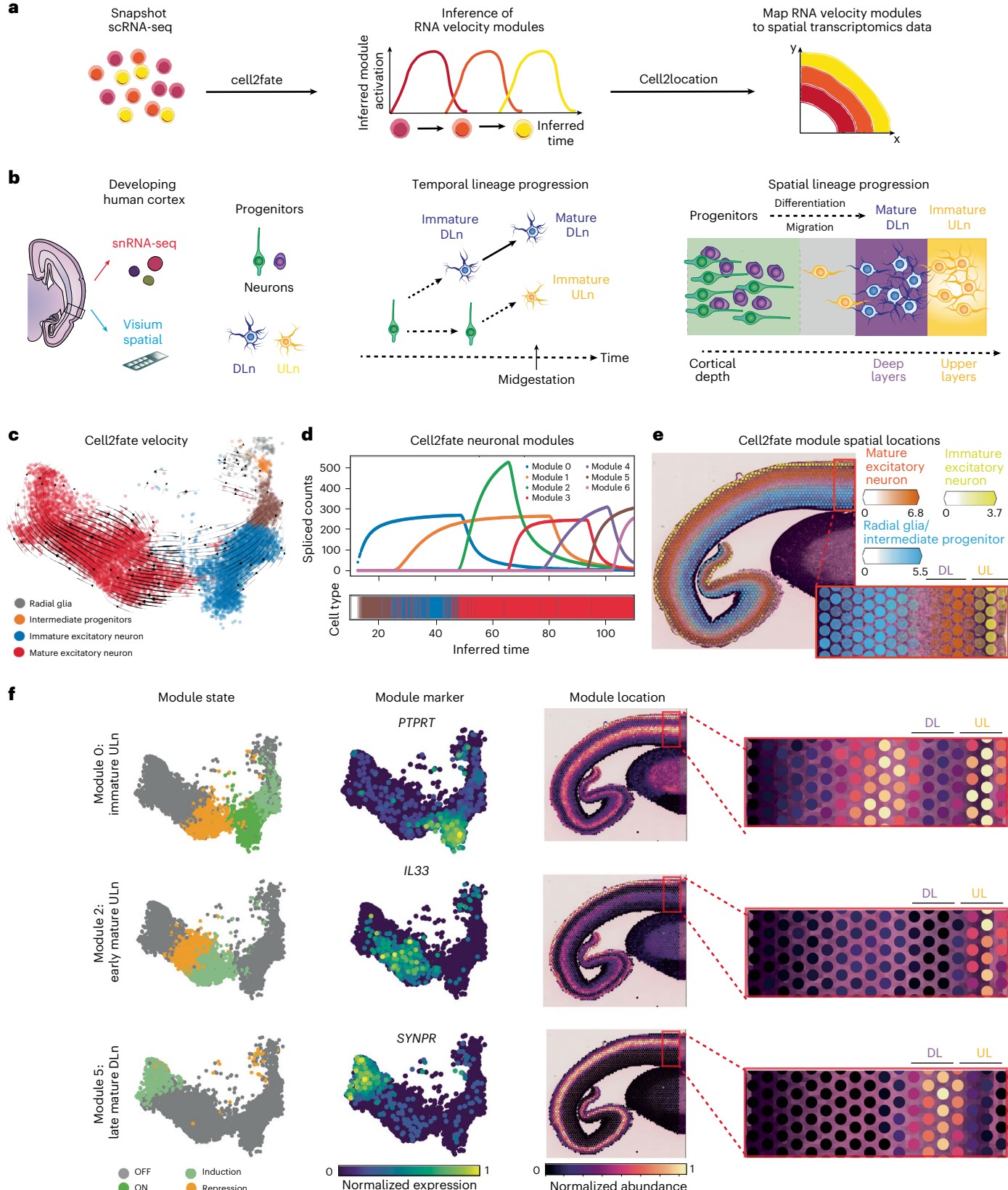

**Fig. 4 | Cell2fate interfaces with cell2location to spatially map the cortical neurogenesis process in human brain development. a**, Cell2fate steady-state expression of modules is supplied as reference profiles for cell2location to infer abundance of modules across spatial locations. **b**, Left, illustration of experimental setup. Middle, radial glia cells (green) and intermediate progenitors (violet) first give rise to deep layer neurons (blue) and then switch to produce ULn (yellow). Right, progenitors (green, violet) are located deeper inside the cortex, where they produce newborn neurons (yellow), which differentiate and migrate toward the outside. **c**, Cell2fate velocity graph UMAP embedding for human brain data. **d**, Spliced counts produced by each cell2fate module over inferred time. **e**, A summary spatial plot of three module locations, named by the cell type in which they reach their steady-state expression. **f**, Module state, markers and locations for individual selected modules. DL, deep layer; UL, upper layer.

well as the subplate–intermediate zone that immature neurons pass through during their migration to the cortical plate[38]. The early mature ULn module (2) was exclusively mapped to the upper cortical layers. By contrast, the late mature DLn module (5) specifically mapped to deep cortical layers.

In sum, our approach provides a workflow to spatially resolve complex cell trajectories through tissues.

## Discussion

Here, we present cell2fate, a Bayesian model of RNA velocity that is capable of inferring transcriptional dynamics in settings of complex changes or weak signals in rare and mature cell types. A core innovation of cell2fate is a formulation of the velocity problem that builds on linearization, which allows for solving a biophysically more accurate model using analytically tractable linearized components. Another benefit of this formulation is that these linear components can be inspected as interpretable RNA velocity modules. This provides for a direct biophysical connection between cell2fate and statistical dimensionality reduction methods. We illustrated this feature by characterizing late maturation trajectories in granule neurons that have been elusive with other methods. Furthermore, RNA velocity modules can be used to locate differentiation trajectories in spatial transcriptomics data. We exemplified this in the developing human brain where the RNA velocity modules of neuronal differentiation showed a high degree of spatial organization.

Despite cell2fate's improved biophysical accuracy, the model still makes simplifying assumptions, such as a constant degradation rate and no stochastic bifurcations. However, the concepts proposed in cell2fate are general and give rise to several extensions that can further increase the biophysical accuracy of the model in the future without resorting to numerical approximations. This includes RNA velocity models with cell-specific splicing and degradation rates, stochastic rates at lineage-branching points and causal connections between transcription rates at different time points, equivalent to dynamic gene regulatory networks (Supplementary Notes). In the long term, dynamical models should also include the effects of cell–cell interactions, based on signaling molecules measured with spatial transcriptomics. An immediate step toward this goal would be combining RNA velocity module mapping with spatial cell–cell interaction tools, such as NCEM[39], which could identify putative interactions that drive specific steps of a differentiation process.

## Online content

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

## Methods

A complete description of the cell2fate model and a comparison with other methods can be found in the Supplementary Notes.

### Generative model for cell2fate

Equations (1) to (3) give rise to these new RNA velocity equations that cell2fate is based on:

$$\frac{d\alpha_g}{dt} = \sum_{m=1}^{M} \frac{d\alpha_{mg}}{dt} = \sum_{m=1}^{M} \lambda_{mi}(\hat{\alpha}_{mgi} - \alpha_{mg}) \qquad (4)$$

$$\frac{du_g}{dt} = \sum_{m=1}^{M} \frac{du_{mg}}{dt} = \sum_{m=1}^{M} (\alpha_{mg} - \beta_g u_{mg}) \qquad (5)$$

$$\frac{ds_g}{dt} = \sum_{m=1}^{M} \frac{ds_{mg}}{dt} = \sum_{m=1}^{M} (\beta_g u_{mg} - \gamma_g s_{mg}). \qquad (6)$$

Subscripts $g$ and $m$ correspond to genes and modules, respectively. $\hat{\alpha}_{mgi}$ is the target transcription rate of each module that is 0 in the OFF state ($i$ = OFF) and nonzero in the ON state ($i$ = ON). The target transcription rate is reached at a rate $\lambda_{mi}$. $\alpha, \beta, \gamma$ denote the transcription, splicing and degradation rates, similar to other RNA velocity models. These RNA velocity equations can be solved analytically (Supplementary Notes 1.3 and 3.1) to give equations for the expected spliced counts $s_{cg}$ and unspliced counts $u_{cg}$ in each cell (Supplementary Note 1.3, equations (15) and (16)). We call these count values 'biological' expectation values, denoted by $x_{cgj}^{B} = (u_{cg}, s_{cg})$, because they represent the expected counts in the absence of measurement effects. The index $j$ denotes maturity of RNA transcripts (spliced/unspliced). To account for measurement effects and obtain 'measurement' expectation values, $x_{cgj}^{M}$, we transform $x_{cgj}^{B}$ as follows:

$$x_{cgj}^{M} = l_{cgj}(H_{ce}s_{egj} + x_{cgj}^{B}). \qquad (7)$$

Here, $H_{ce}$ denotes a one-hot categorical assignment of cells to experimental batches, $l_{cgj}$ describes differences in detection efficiency (for example, sequencing depth, read alignment and quantification) of genes across cells and $s_{egj}$ models ambient RNA ('soup') for each gene in each batch. These 'measurement' expectation values are then used to parameterize the mean of a negative binomial (NB) observation model of the observed raw count values $X_{cgj} = (U_{cg}, S_{cg})$:

$$X_{cgj} \sim \text{NB}\left(\mu = x_{cgj}^{M}, \alpha = a_{gj}\right). \qquad (8)$$

Here, $a_{gj}$ are NB overdispersion parameters for each gene, separate for spliced and unspliced counts. All parameters have hierarchical prior distributions (Supplementary Note 1.6) and are inferred using stochastic variational inference in the Pyro probabilistic programming framework[40] (Supplementary Note 1.8).

### Downstream analysis of cell2fate

**Computation of the RNA velocity graph.** We followed the procedure proposed in the scVelo package for computing a cell–cell transition probability graph from RNA velocity estimates[2], with the modification of averaging the velocity graph over 100 posterior velocity samples, so that noisy gene velocities with high posterior uncertainty have less weight in estimating transition probabilities. Optionally, the original procedure can also be followed exactly by computing the velocity graph with mean velocity estimates from our method, using the original function in the scVelo Python package.

**Computation of module activation.** We calculate the module activation, which we defined as the total spliced counts produced by a module in a cell, by substituting posterior parameter values into the equations that capture the time evolution of spliced counts. Module activation can then be plotted over time by plotting the posterior time of each cell on the $x$ axis and the calculated module activation of each module on the $y$ axis.

**Calculation of normalized module activation and state.** Module normalized activation denotes the number of counts produced by a module divided by the steady-state counts of the module (Fig. 3d, gray line), which is calculated by setting time to infinity in the equations that capture the time evolution of spliced counts. The module state is defined as OFF if the cell time is smaller than the switch-on time of the module or the module's normalized activation is below 0.05 (Fig. 3d, gray) and as ON if its normalized activation is above 0.95 (Fig. 3d, bright green). Otherwise, a module is in either the induction or repression state depending on whether the inferred cell time is below or above the switch-off time (Fig. 3d, light green or orange).

### Benchmarking of RNA velocity methods

**Processing of datasets.** We used the 3,000 most variable genes with at least 20 detected counts for all results in the paper. In addition, we applied the standard preprocessing pipeline suggested in the main scVelo tutorial (on mouse pancreas development), except for Pyro-Velocity and cell2fate, which do not require preprocessing. This included total count normalization, log transformation and calculating mean expression among the 30 nearest neighbors in 30-component PCA space ('kNN-smoothing').

### Application of velocity models

We followed online tutorials based on the mouse pancreas development dataset for all methods and then used the same analysis pipeline to produce the benchmarking results in this paper. We summarize parameter settings for all methods in Table 1 and also include the method version when available. For cell2fate, we kept all training and model parameters at their default values.

**Definition of ground truth cell state transitions.** For the mouse bone marrow dataset, we defined the following ground truth cluster transitions: 'dividing' to 'progenitors' to 'activating'. For the remaining datasets, we used the ground truth transitions from the UniTVelo RNA velocity study that also used these datasets for benchmarking[41]. All ground truth transitions are included in Supplementary Tables 1–5.

**Benchmarking metric.** We calculated the CBDir, using the functions provided by the UniTVelo Python package[41]. The following is an explanation of the metric from the UniTVelo publication: 'CBDir measures the correctness of transitions from a source cluster to target cluster using boundary cells given ground truth. Here boundary of source cluster refers to cells in that cluster that are neighbors of the target cluster and vice versa. Boundary cells are used because they reflects the biological development in a short period of time and CBDir is calculated via

$$\text{CBDir}(c) = \frac{1}{\text{Norm}} \sum_{c' \in C_A \cap N(c)} \frac{v_c \cdot (x_{c'} - x_c)}{|v_c| \cdot |x_{c'} - x_c|}$$

$$\text{Norm} = |c' \epsilon C_A \cap N(c)|, \qquad (9)$$

where $C_A$ is sets of cells in target cluster $A$, $N(c)$ stands for the neighboring cells of specified cell $c$. $v_c$ and $x_c$ are the low-dimensional vectors representing computed velocity and positions of cell $c$. Therefore, $x_{c'} - x_c$ is the displacement in space during the short period of time'.

**CellRank analysis.** For the dentate gyrus dataset, we ran CellRank with default parameters and computed fate probabilities for the astrocyte, oligodendrocyte and granule mature states, if the method identified them as macrostates. When using the default parameter

n_states = (4, 12), VeloVI and VeloVAE returned an error: 'ValueError: Clustering into 12 clusters will split complex conjugate eigenvalues. Request one cluster more or less'; so we set n_states = (4, 11), which ran successfully. For cellDancer and VeloVAE, we also obtained the following error: '[…] value(s) do not sum to 1 (rtol=1e-3). Try decreasing the tolerance as 'tol = …', specifying a preconditioner as 'preconditioner = …' or use a direct solver as 'solver = 'direct'' if the matrix is small'. This error could not be resolved, even after following the steps in the error message and the advice found regarding relevant GitHub issues. For the erythroid maturation dataset, we also ran CellRank with default parameters and computed fate probabilities both toward the progenitor 1 state and the erythroid 3 state.

**Simulation benchmark.** First, we fit cell2fate to the dentate gyrus dataset. Next, we generated data from the cell2fate generative model, keeping all parameters at their fitted values except one chosen parameter that we multiplied with 0.25, 0.5, 1, 2 or 4. We investigated four biological and technical parameters in this way: the splicing rate, the degradation rate, the detection probability of transcripts and the NB overdispersion parameter. We then fit cell2fate to this simulated data and calculated the correlation of inferred splicing and degradation rate values to the known ground truth (as RNA velocity is a function of splicing and degradation rates).

**Time and memory requirements.** We measured the time requirements for all methods using the 'time' Python package. The start point for time measurements was placed right before data preprocessing began, and the end point was placed right after velocity values were obtained. We measured memory requirements by running either the 'htop' (for CPU-based methods) or the 'nvidia-smi' command (for GPU-based methods) every 10 s for up to 2 min and recording the highest value.

### Comparison of decomposition methods on the dentate gyrus dataset

**Application of cell2fate.** Following the benchmarking run on dentate gyrus data, we applied the downstream analysis methods, described above in Downstream analysis, to calculate and plot module activations and states.

**Application of multi-omics factor analysis.** We used the 3,000 most variable genes with at least 20 detected counts as input, identical to the cell2fate analysis. We limited the analysis to clusters involved in the neuron differentiation trajectory (radial glia like, nIPC, neuroblast, granule immature, granule mature). We added spliced and unspliced counts and then normalized, log transformed and scaled the data matrix using the respective scanpy functions. We ran MOFA using a Gaussian likelihood and ten factors (the same number of factors found by cell2fate in the relevant clusters). Further run options were spikeslab weights = True, ard factors = True, ard weights = True.

**ProBound algorithm.** To produce Supplementary Fig. 29, we applied the ProBound algorithm[33]. ProBound can predict the binding affinity for most TFs to a user-supplied DNA sequence. We used the refdata-cellranger-arc-mm10-2020-A-2.0.0 genome as a reference. To obtain putative promoter sequences for each gene in the reference, we extracted the DNA sequence 450 bp upstream and 149 bp downstream of the transcription start site of each gene. For each TF, we then ranked genes based on the predicted ProBound binding affinity to its putative transcription start site. We then used the top ten and bottom ten binding targets of each TF to produce the results illustrated in Supplementary Fig. 29.

### Spatial integration of RNA velocity

**Human tissue.** Formalin-fixed paraffin-embedded (FFPE) blocks of second-trimester human fetal brain were obtained from the MRC- and Wellcome-funded Human Developmental Biology Resource (HDBR, http://www.hdbr.org), with appropriate maternal written consent and approval from the Fulham Research Ethics Committee (REC reference 18/LO/0822) and the Newcastle & North Tyneside 1 Research Ethics Committee (REC reference 18/NE/0290). The HDBR is regulated by the UK Human Tissue Authority (http://www.hta.gov.uk) and operates in accordance with the relevant Human Tissue Authority Codes of Practice.

**Developing human brain single-nucleus library preparation and sequencing.** Single nuclei were isolated from frozen fetal brain tissue according to a published protocol[42]. Briefly, tissue was Dounce homogenized in homogenization buffer (250 mM sucrose, 25 mM potassium chloride, 5 mM magnesium chloride, 10 mM Tris buffer, pH 8.0, 1 M 1,4-dithiothreitol, 0.1% Triton X-100, 1× protease inhibitor, 0.4 U l$^{-1}$ RNasin Plus RNase inhibitor, 0.2 U l$^{-1}$ SUPERase·In RNase inhibitor) and filtered with a 40-µm cell strainer. Debris was removed from the filtrate via density centrifugation with 27% Percoll. All nuclei in a batch were mixed in equal concentrations before droplet encapsulation with the 10x Chromium Single Cell 3′ version 3.1 kit. Libraries were generated according to the manufacturer's protocol (CG000204) and single-index sequenced with cycles 28-8-91 on a NovaSeq 6000 System (Illumina) using a NovaSeq S4 flow cell.

**Developing human brain Visium library preparation and sequencing.** FFPE tissue sections (5 µm) were stained and imaged following the 10x Genomics Visium CytAssist user guide (CG000520). The following times were used for the fetal brain tissue: hematoxylin, 3 min; bluing, 1 min; eosin, 1 min. After probe hybridization and ligation, a Visium CytAssist instrument was used to transfer analytes from the glass slide to a Visium CytAssist Spatial Gene Expression slide with a capture area of 42.25 mm$^2$. Probe extension and library construction were carried out following the standard Visium for FFPE workflow (CG000495) outside of the instrument. Libraries were sequenced with paired-end dual indexing (28 cycles, read 1; ten cycles, i7; ten cycles, i5; 90 cycles, read 2) on the Illumina-HTP NovaSeq 6000 System with paired-end sequencing in an SP flow cell. The Loupe Browser was used to generate the JSON file, and the Space Ranger pipeline version 2022.0705.1 (10x Genomics) and the GRCh38-2020-A reference were used to process FASTQ files.

**Developing human brain Visium and single-nucleus sequencing count quantification, clustering and annotation.** We quantified counts using Space Ranger 1.3.0 for the Visium data and STARsolo with Cell Ranger 3.02 for the single-nucleus data using GRCh38 version 1.2.0 as a reference. We applied CellBender to the single-nucleus total count matrix (but not spliced and unspliced count matrices). We followed the scanpy processing and clustering tutorial with default parameters (min_genes = 200, min_cells = 3, n_genes_by_counts < 2,500, pct_counts mt < 5), which involves removing cells and genes with low UMI counts, followed by removal of cells with very high total counts or mitochondrial gene ratio, total count normalization, log transformation, highly variable gene selection, data scaling, principal-component analysis and finally Louvain clustering. Expression of cell type marker genes taken from the single-cell atlas (Fig. 1f of Polioudakis et al.[35]) was plotted for each cluster, based on which we annotated the cluster identity. Clusters outside the excitatory neuron lineage (oligodendrocytes, interneurons, microglia) were not considered for further analysis.

**Cell2location.** We used the steady-state counts of each module as a reference gene expression profile. This steady-state expression corresponds to the $g_{mg}$ parameter of the generative model. We then ran the cell2location method with default parameter settings (including $\alpha = 20$).

**Reporting summary**

Further information on research design is available in the Nature Portfolio Reporting Summary linked to this article.

## Data availability

Raw UMI counts and metadata in anndata format for all single-cell and Visium data are available for download on this portal: https://cell2fate.cog.sanger.ac.uk/browser.html. FASTQ files for the human brain single-nucleus and Visium data are available on ENA under the accession number PRJEB79988.

## Code availability

The cell2fate package is available for installation at this GitHub repository: https://github.com/BayraktarLab/cell2fate. It is also available on Zenodo[40]: https://zenodo.org/records/13883214. Results from the cell2fate method can be reproduced with the notebooks in this repository: https://github.com/AlexanderAivazidis/cell2fate_notebooks. Benchmarking results for all methods as well as robustness analysis and comparison to real developmental age were done with notebooks in this repository: https://github.com/AlexanderAivazidis/fate_benchmarking.

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

## Acknowledgements

We gratefully acknowledge L. Parts, Y. Huang, M. Gao and C. Qiao for valuable discussions on the cell2fate model and E. Prigmore and J.E. Kwa for assistance with human brain single-nucleus transcriptomics. This work was supported by Wellcome Leap as part of the Delta Tissue program to O.A.B. and O.S., European Commission (ERC project DECODE, 810296) funding to O.S. and Wellcome Trust grants 206194 and 220540/Z/20/A to O.A.B. The human embryonic and fetal material was provided by the Joint MRC–Wellcome Trust (grant MR/006237/1) HDBR (http://www.hdbr.org).

## Author contributions

A.A. conceived the cell2fate model, implemented and tested it and produced all figures and results in the paper. F.M. generated the human brain single-nucleus and Visium data. V.K. contributed to model conception and implementation in Pyro. S.E. added unit tests and documentation and contributed to improvements in the Pyro implementation. B.C. contributed to model conception, supervision and paper editing. O.S. and O.A.B. supervised A.A. A.A., O.S. and O.A.B. co-wrote the paper with feedback from all authors.

## Competing interests

O.S. is a paid advisor of insitro. B.C. holds equity in Tachyon Therapeutics. The other authors declare no competing interests.

## Additional information

**Correspondence and requests for materials** should be addressed to Oliver Stegle or Omer Ali Bayraktar.

# Reporting Summary

## Statistics

For all statistical analyses, confirm that the following items are present in the figure legend, table legend, main text, or Methods section.

| n/a | Confirmed | |
|---|---|---|
| ☐ | ☒ | The exact sample size (*n*) for each experimental group/condition, given as a discrete number and unit of measurement |
| ☐ | ☒ | A statement on whether measurements were taken from distinct samples or whether the same sample was measured repeatedly |
| ☒ | ☐ | The statistical test(s) used AND whether they are one- or two-sided *Only common tests should be described solely by name; describe more complex techniques in the Methods section.* |
| ☒ | ☐ | A description of all covariates tested |
| ☐ | ☒ | A description of any assumptions or corrections, such as tests of normality and adjustment for multiple comparisons |
| ☐ | ☒ | A full description of the statistical parameters including central tendency (e.g. means) or other basic estimates (e.g. regression coefficient) AND variation (e.g. standard deviation) or associated estimates of uncertainty (e.g. confidence intervals) |
| ☒ | ☐ | For null hypothesis testing, the test statistic (e.g. *F*, *t*, *r*) with confidence intervals, effect sizes, degrees of freedom and *P* value noted *Give P values as exact values whenever suitable.* |
| ☐ | ☒ | For Bayesian analysis, information on the choice of priors and Markov chain Monte Carlo settings |
| ☒ | ☐ | For hierarchical and complex designs, identification of the appropriate level for tests and full reporting of outcomes |
| ☐ | ☒ | Estimates of effect sizes (e.g. Cohen's *d*, Pearson's *r*), indicating how they were calculated |

*Our web collection on statistics for biologists contains articles on many of the points above.*

## Software and code

Policy information about availability of computer code

| Data collection | The single-nucleus RNAseq data was processed using the open-source StarSolo method, version 2.7.9a with the velocyto option enabled. Available at: https://github.com/alexdobin/STAR/blob/master/docs/STARsolo.md |
|---|---|
| Data analysis | Analysis of RNAseq data was performed with the cell2fate method available here: https://github.com/BayraktarLab/cell2fate |
| | Results from the cell2fate method can be reproduced with the notebooks in this repository: github.com/AlexanderAivazidis/cell2fate_notebooks |
| | Benchmarking results for all methods, as well as robustness analysis and comparison to real developmental age was done with notebooks in this repository: github.com/AlexanderAivazidis/fate_benchmarking |
| | We used a computing environment with the following publically available python packages: |

```
Name            Version
_libgcc_mutex      0.1
_openmp_mutex       4.5
absl-py           1.2.0
aiohttp           3.8.1
aiosignal         1.2.0
anndata           0.8.0
appdirs           1.4.4
```

```
argon2-cffi              21.3.0
argon2-cffi-bindings     21.2.0
arpack                   3.7.0
asttokens                2.0.5
astunparse               1.6.3
async-timeout            4.0.2
attrs                    22.1.0
backcall                 0.2.0
beautifulsoup4           4.11.1
bioservices              1.10.0
blas                     1.0
bleach                   5.0.1
bottleneck               1.3.5
brotlipy                 0.7.0
c-ares                   1.18.1
ca-certificates          2022.9.14
cachetools               5.2.0
cell2fate                0.1a0
cell2location            0.1
certifi                  2022.9.14
cffi                     1.15.1
charset-normalizer       2.1.0
chex                     0.1.4
click                    8.1.3
colorama                 0.4.5
colorlog                 6.6.0
commonmark               0.9.1
cryptography             35.0.0
cycler                   0.11.0
debugpy                  1.6.2
decorator                5.1.1
defusedxml               0.7.1
dm-tree                  0.1.7
docrep                   0.3.2
easydev                  0.12.0
einops                   0.4.1
entrypoints              0.4
et-xmlfile               1.1.0
etils                    0.6.0
executing                0.9.1
fastcluster              1.2.6
fastjsonschema           2.16.1
flatbuffers              22.12.6
flax                     0.5.0
fonttools                4.34.4
freetype                 2.10.4
frozenlist               1.3.1
fsspec                   2022.7.1
future                   0.18.2
gast                     0.4.0
gevent                   21.8.0
glpk                     4.65
gmp                      6.2.1
google-auth              2.9.1
google-auth-oauthlib     0.4.6
google-pasta             0.2.0
greenlet                 1.1.1
grequests                0.6.0
grpcio                   1.47.0
gseapy                   0.12.1
h5py                     3.7.0
html5lib                 1.1
icu                      58.2
idna                     3.3
igraph                   0.9.10
importlib-metadata       3.10.0
importlib-resources      5.9.0
intel-openmp             2021.4.0
iprogress                0.4
ipykernel                6.15.1
ipython                  8.4.0
ipython-genutils         0.2.0
ipywidgets               7.7.1
jax                      0.3.15
jaxlib                   0.3.15
jedi                     0.18.1
```

```
jinja2               3.1.2
joblib               1.1.0
jpeg                 9e
jsonschema           4.9.1
jupyter              1.0.0
jupyter-client       7.3.4
jupyter-console      6.4.4
jupyter-core         4.11.1
jupyterlab-pygments  0.2.2
jupyterlab-widgets   1.1.1
keras                2.11.0
kiwisolver           1.4.4
lcms2                2.12
ld_impl_linux-64     2.38
leidenalg            0.8.10
libblas              3.9.0
libcblas             3.9.0
libclang             14.0.6
libev                4.33
libffi               3.3
libgcc-ng            12.1.0
libgfortran-ng       7.5.0
libgfortran4         7.5.0
libhwloc             2.8.0
liblapack            3.9.0
libpng               1.6.37
libstdcxx-ng         12.1.0
libtiff              4.2.0
libuv                1.40.0
libwebp-base         1.2.2
libxml2              2.9.14
libxslt              1.1.35
libzlib              1.2.12
llvm-openmp          14.0.4
llvmlite             0.39.0
loompy               3.0.7
lxml                 4.9.1
lz4-c                1.9.3
markdown             3.3.4
markupsafe           2.1.1
matplotlib           3.5.2
matplotlib-base      3.4.3
matplotlib-inline    0.1.3
metis                5.1.0
mistune              0.8.4
mkl                  2021.4.0
mkl-service          2.4.0
mkl_fft              1.3.1
mkl_random           1.2.2
mpfr                 4.1.0
msgpack              1.0.4
multidict            6.0.2
multipledispatch     0.6.0
natsort              8.1.0
nbclient             0.6.6
nbconvert            6.5.0
nbformat             5.4.0
ncurses              6.3
nest-asyncio         1.5.5
networkx             2.8.5
notebook             6.4.12
numba                0.56.0
numexpr              2.8.3
numpy                1.21.4
numpy-groupies       0.9.17
numpyro              0.10.0
oauthlib             3.2.0
olefile              0.46
opencv-python        4.6.0.66
openpyxl             3.0.10
openssl              1.1.1q
opt-einsum           3.3.0
optax                0.1.3
packaging            21.3
pandas               1.4.2
pandocfilters        1.5.0
```

```
parso                            0.8.3
patsy                            0.5.2
pexpect                          4.8.0
pickleshare                      0.7.5
pillow                           9.2.0
pip                              22.1.2
prometheus-client                0.14.1
prompt-toolkit                   3.0.30
protobuf                         3.19.6
psutil                           5.9.1
ptyprocess                       0.7.0
pure-eval                        0.2.2
pyasn1                           0.4.8
pyasn1-modules                   0.2.8
pycparser                        2.21
pydeprecate                      0.3.1
pygments                         2.12.0
pynndescent                      0.5.7
pyopenssl                        22.0.0
pyparsing                        3.0.9
pyro-api                         0.1.2
pyro-ppl                         1.8.1
pyrsistent                       0.18.1
pysocks                          1.7.1
python                           3.9.12
python-dateutil                  2.8.2
python-graphviz                  0.20.1
python-igraph                    0.9.11
python_abi                       3.9
pytorch-lightning                1.5.10
pytz                             2022.1
pyyaml                           6.0
pyzmq                            23.2.0
qtconsole                        5.4.0
qtpy                             2.3.0
readline                         8.1.2
requests                         2.28.1
requests-oauthlib                1.3.1
requests_cache                   0.4.13
rich                             12.3.0
rsa                              4.9
scanpy                           1.9.1
scikit-learn                     1.1.1
scipy                            1.8.0
scvelo                           0.2.4
scvi-tools                       0.16.1
seaborn                          0.11.2
send2trash                       1.8.0
session-info                     1.0.0
setuptools                       59.5.0
six                              1.16.0
soupsieve                        2.3.2.post1
sqlite                           3.39.0
stack-data                       0.3.0
statsmodels                      0.13.2
stdlib-list                      0.8.0
suds-community                   1.1.2
suitesparse                      5.10.1
tbb                              2021.5.0
tensorboard                      2.11.0
tensorboard-data-server          0.6.1
tensorboard-plugin-wit           1.8.1
tensorflow                       2.11.0
tensorflow-estimator             2.11.0
tensorflow-io-gcs-filesystem     0.28.0
termcolor                        2.1.1
terminado                        0.15.0
texttable                        1.6.4
threadpoolctl                    3.1.0
tinycss2                         1.1.1
tk                               8.6.12
toolz                            0.12.0
torch                            1.11.0
torchmetrics                     0.9.3
tornado                          6.2
tqdm                             4.64.0
```

```
traitlets          5.3.0
txnburst           0.0.0
typing-extensions  4.3.0
tzdata             2022a
umap-learn         0.5.3
unitvelo           0.2.5
urllib3            1.26.11
wcwidth            0.2.5
webencodings       0.5.1
werkzeug           2.2.1
wheel              0.37.1
widgetsnbextension 3.6.1
wrapt              1.14.1
xmltodict          0.13.0
xz                 5.2.5
yarl               1.8.1
zipp               3.8.1
zlib               1.2.12
zope.event         4.5.0
zope.interface     5.4.0
zstd               1.4.9
```

Additional algorithms run in our study are listed in the following together with their version number:

scvelo_dynamical
v0.2.5
scvelo_stochastic
v0.2.5
pyroVelocity_model1
0.1.0
veloVAE
n.a.
UniTVelo_independent
0.2.5
UniTVelo_unified
0.2.5
DeepVelo
0.2.5-rc.1
pyroVelocity_model2
0.1.0
VeloVI
0.1.1
CellRank
v.2.0.6
MOFA
v0.1
ProBound
1.4.0

For manuscripts utilizing custom algorithms or software that are central to the research but not yet described in published literature, software must be made available to editors and reviewers. We strongly encourage code deposition in a community repository (e.g. GitHub). See the Nature Portfolio guidelines for submitting code & software for further information.

# Data

Policy information about availability of data

All manuscripts must include a data availability statement. This statement should provide the following information, where applicable:

- Accession codes, unique identifiers, or web links for publicly available datasets
- A description of any restrictions on data availability
- For clinical datasets or third party data, please ensure that the statement adheres to our policy

Raw UMI counts and metadata in anndata format for all single cell and Visium data is available for download on this portal: https://cell2fate.cog.sanger.ac.uk/browser.html
FASTQ files for the human brain single-nucleus and Visium data are available on ENA under this accession number: PRJEB79988.
Reference GRCh38 v1.2.0 that was used to process the FASTQ files is available for download here: https://www.ncbi.nlm.nih.gov/datasets/genome/GCF_000001405.26/

## Human research participants

Policy information about studies involving human research participants and Sex and Gender in Research.

| | |
|---|---|
| Reporting on sex and gender | The sex was assigned as male by the Human Developmental Biology Resource, UCL, UK (REC 23/LO/0312) from where the second trimester human fetal brain tissue was obtained. |
| Population characteristics | No further characteristics were provided. |
| Recruitment | Human embryo and fetal samples were obtained from the MRC and Wellcome-funded Human Developmental Biology Resource (HDBR, http:// www.hdbr.org), with appropriate maternal written consent and approval from the Fulham Research Ethics Committee (REC reference 18/LO/0822) and Newcastle & North Tyneside 1 Research Ethics Committee (REC reference 18/NE/0290). The HDBR is regulated by the UK Human Tissue Authority (HTA; www.hta.gov.uk) and operates in accordance with the relevant HTA Codes of Practice. |
| Ethics oversight | Human embryo and fetal samples were obtained from the MRC and Wellcome-funded Human Developmental Biology Resource (HDBR43, http:// www.hdbr.org), with appropriate maternal written consent and approval from the Newcastle and North Tyneside NHS Health Authority Joint Ethics Committee (08/H0906/21+5). The HDBR is regulated by the UK Human Tissue Authority (HTA; www.hta.gov.uk) and operates in accordance with the relevant HTA Codes of Practice. |

Note that full information on the approval of the study protocol must also be provided in the manuscript.

# Field-specific reporting

Please select the one below that is the best fit for your research. If you are not sure, read the appropriate sections before making your selection.

☒ Life sciences  ☐ Behavioural & social sciences  ☐ Ecological, evolutionary & environmental sciences

For a reference copy of the document with all sections, see nature.com/documents/nr-reporting-summary-flat.pdf

# Life sciences study design

All studies must disclose on these points even when the disclosure is negative.

| | |
|---|---|
| Sample size | No sample size calculation was performed, since we did not use a case control study design, so there were no comparisons between groups. Instead we demonstrated our computational method on 5 public and 1 new dataset, for which Visium spatial transcriptomics and single-nucleus RNA sequencing were applied to one tissue sample from the same donor. |
| Data exclusions | Data was excluded during quality control of the single-nucleus RNAseq data, using default recommended count thresholds in the scanpy python processing pipeline: min_genes=200, min_cells=3, n_genes_by_counts = 2500, pct_counts_mt = 5 |
| Replication | We did not replicate any experiments, since this is not generally needed for single-cell RNA sequencing, which is a very reproducible assay. |
| Randomization | There are no comparisons between experimental groups in our study hence no need for randomization. |
| Blinding | There are no comparisons between experimental groups in our study hence no need for blinding. |

# Reporting for specific materials, systems and methods

We require information from authors about some types of materials, experimental systems and methods used in many studies. Here, indicate whether each material, system or method listed is relevant to your study. If you are not sure if a list item applies to your research, read the appropriate section before selecting a response.

### Materials & experimental systems

| n/a | Involved in the study |
|---|---|
| ☒ ☐ | Antibodies |
| ☒ ☐ | Eukaryotic cell lines |
| ☒ ☐ | Palaeontology and archaeology |
| ☒ ☐ | Animals and other organisms |
| ☒ ☐ | Clinical data |
| ☒ ☐ | Dual use research of concern |

### Methods

| n/a | Involved in the study |
|---|---|
| ☒ ☐ | ChIP-seq |
| ☒ ☐ | Flow cytometry |
| ☒ ☐ | MRI-based neuroimaging |

