## [Peer Review File · Nature Methods]

Cell2fate infers RNA velocity modules to improve cell fate prediction

Corresponding Author: Dr Omer Ali Bayraktar

Version 0:

Decision Letter:

12th Jan 2024

Dear Dr Bayraktar,

Your Article, "Model-based inference of RNA velocity modules improves cell fate prediction", has now been seen by 3 reviewers. As you will see from their comments below, although the reviewers find your work of potential interest, they have raised a number of concerns. We are interested in the possibility of publishing your paper in Nature Methods, but would like to consider your response to these concerns before we reach a final decision on publication.

We therefore invite you to revise your manuscript to address these concerns. Addressing Reviewer 3's Major Concern 4 about generating new biological insights is encouraged but not a must.

Link Redacted

We hope to receive your revised paper within 3 months. If you cannot send it within this time, please let us know. In this event, we will still be happy to reconsider your paper at a later date so long as nothing similar has been accepted for publication at Nature Methods or published elsewhere.

OPEN SCIENCE REQUIREMENTS

REPORTING SUMMARY AND EDITORIAL POLICY CHECKLISTS

DATA AVAILABILITY

All novel DNA and RNA sequencing data, protein sequences, genetic polymorphisms, linked genotype and phenotype data, gene expression data, macromolecular structures, and proteomics data must be deposited in a publicly accessible database, and accession codes and associated hyperlinks must be provided in the "Data Availability" section.

CODE AVAILABILITY

Please include a "Code Availability" subsection in the Online Methods which details how your custom code is made available. Only in rare cases (where code is not central to the main conclusions of the paper) is the statement "available upon request" allowed (and reasons should be specified).

MATERIALS AVAILABILITY

ORCID

Sincerely,

Lin Tang, PhD
Senior Editor
Nature Methods

Reviewers' Comments:

Reviewer #1:

Remarks to the Author:

Summary

The authors present cell2fate, a Bayesian model to estimate RNA velocity in a modular fashion. Their main innovation is to parametrize the transcription rate as a sum over different modules, which induces a modular structure for spliced and unspliced counts as well. With this formulation, the ODEs for each module can be solved analytically, and model parameters can be fit using stochastic variational inference. Their formulation of RNA velocity offers the following advantages:

- Operating on the level of gene modules makes a lot of sense. We know that genes act in modules for many biological processes. This allows for more flexible dynamical patterns of the transcription rate, while remaining computationally tractable. Framing the problem this way is both biologically as well as numerically a useful approximation.
- Operating directly on the level of raw mRNA counts, avoiding imputation.
- Explicitly accounting for technical effects, such as batch effects, ambient mRNA, and detection sensitivity of spliced/unspliced mRNA.
- Fitting in a fully Bayesian fashion, which allows for the inclusion of useful information via priors (e.g. sparsity, effective number of active modules, etc.). Also, this allows the authors to estimate velocity and parameter uncertainty.

Their method is a useful addition to the suite of RNA velocity models, and shows promising performance in a number of applications and benchmarks. The idea of combining velocity inference with a membership-like model is innovative and offers several avenues for future improvements, as the authors lie out in the discussion. I think this work will be of broad interest to biologists and bioinformaticians working with single-cell data from various dynamical processes.

I have a number of concerns relating to the metrics used for benchmarking, visual inspection of velocity streams, manual definition of interesting trajectories, method scalability, and the choice of datasets (see my major comments below). Most importantly, it would be great to see cell2fate's performance in a more challenging real-data setting, like regeneration, reprogramming or cancer, where simpler approaches fail.

Major comments

1. Fig. 2, benchmarking of velocity methods. I appreciate the authors efforts to compare 10 different velocity models across 5 different datasets, but I'm not fully satisfied with their metrics. The choice CDBir seems appropriate, yet I would encourage them to include another quantitative measure to evaluate their trajectories. For example, the authors could work more with their cell-specific latent time, and check whether it correctly increases with increasing maturity. This could be done in at least two ways: (1) for datasets that contain experimental time points, they could check whether their latent time increases from early to late time points, (2), for single-snapshot datasets, they could define a very coarse ground-truth ordering of cell populations/types (which should be easy e.g. in hematopoiesis), and compare the temporal ordering indicated by their latent time with ground truth, using e.g. spearman rank correlation between the ground truth ordering and the mean latent time per cell type. Their current panel E goes in this direction, but does not include any quantitative evaluation.
2. Projected velocity streams over-interpretation: I would also urge the authors to rely less on projected velocity streams in Panels B and C of Fig. 2, as these can be very misleading, as noted in several previous publications. The same holds for a few other panels throughout the manuscript (e.g. SFig. 22 and others). I think it would be better to evaluate the resulting trajectories quantitatively in high dimensions, using tools like CellRank or dynamo. Especially for small populations like the OPCs, I think projected streamplot on a UMAP convey very little information and are prone to over-interpretation. For example, L215, "(...) while radial glial-like progenitor cells are largely committed to astrocytes", if that statement has been derived from eye-balling the UMAP in Fig. 3A, then I would strongly advice for using a more quantitative approach of estimating fate probabilities.
3. Defining trajectories: In Fig. 3, it appears as if the authors had to manually subset their data to characterize the granule

maturation trajectory. The same appears to be the case in Fig. 4 for the excitatory neuronal lineage. Could this be automated by e.g. using a tool like CellRank or dynamo to compute fate probabilities, and then select those cells which are likely to become granule neurons (as opposed to Astrocytes)? Computing fate probabilities to differentiate between different trajectories in an unbiased fashion is an important use case of RNA velocity.

4. Method scalability, see Methods, Fig. 2. I expect this model to be computationally more expensive compared to scVelo's dynamical model, especially when using parallelization for scVelo. On a dataset of 10k cells across 3k genes, on a modern MacBook with 8 cores, I expect scVelo's dynamical model to finish in around 7-10 minutes. It would be good to have some sort of comparison here (in terms of runtime and memory consumption) with a baseline model to illustrate the trade off between the additional complexity of cell2fate, and increased computational cost. Also, it would be good to investigate how compute time scales with the number of cells, modules and genes, either theoretically or through experiments. I suppose this model is currently geared towards small-scale single cell datasets containing less than 50k cells (the largest dataset in this study is below 10k cells). It would be good to state this as a limitation in the discussion, if that's the case, and to discuss how future improvements could make the model more scalable and applicable to atlas-sized datasets. In principle, batch effect correction is a very valuable component to achieve this goal, so cell2fate could be well positioned if it achieves the necessary scalability.

5. Datasets. Currently, cell2fate has been demonstrated on a number of developmental (mouse pancreas, dentate gyrus, and erythroid differentiation and human neurogenesis) and steady-state (mouse and human bone marrow) processes. However, these are arguably processes where simpler methods, like pseudotime inference, work well and provide many biological insights as well. I would be interested to see the performance of cell2fate in a more challenging setting where simpler approaches, like PAGA, slingshot, Palantir, or DPT, fail. Settings like regeneration or cancer are usually more challenging, because we don't know the correct directions a priori, and we cannot just subset to the trajectory we want to study (because that's what we need to discover). These are the settings where RNA velocity is most promising as a concept, and this is where cell2fate should provide new insights that we could not get with competing velocity approaches, or simpler pseudotime approaches.

Minor comments

L46: What do the authors refer to with "unconstrained" when they say "(...) that allows for capturing unconstrained transcriptional dynamics while retaining computational and numerical tractability." ?

L50: The following sentence is unclear to me: "Cell2fate employs a linearization to decompose complex differential equations into tractable components that can be solved analytically". The 2-ODE formulation of RNA velocity, which corresponds to a first-order moment approximation of the Chemical Master Equation (CME), consists of 2 linear, first-order ODEs, which can be solved analytically if we assume constant rates (and one switch for α). The authors should be more explicit at this point about what they mean when they say "complex differential equations." Why are they complex? One problem is clearly that we don't measure time explicitly, but that's a problem with the data, not with the ODE system.

L63, "Cell2fate builds on a long-standing history of computational methods for RNA velocity (...)" is an overstatement. RNA velocity for single cells has existed since 2018, that's not really a "long history."

L224, "This analysis identified the successive induction of modules across the early differentiation of radial glia into nIPCs, neuroblasts and immature neurons", I thought you found radial glial cells to be mostly committed to Astrocytes here?

Fig. 3, I think the legend description for panels B and C is mixed up.

Fig. 3, does the data displayed in Panel B correspond only to the subset of cell highlighted with the red box in A? This is unclear.

Fig. 3D, I don't really understand this panel. This could be explained better in the Figure and text

Additional minor comments on the Methods

Methods, Eq (1-3): "(...) these differential equations have been suggested (...)". There is a clear link between Eq (1) and Eqs (2-3). The latter two equations correspond to the first-order moment approximation of the CME corresponding to Eq (1).

Methods L88, describe the problems with parameter fitting here briefly.

Methods, L101, is that also true for the scVelo model? I thought they could somehow be pooled post-hoc to live on the same scale.

Methods, L164, which is there a transpose here? Is that not a scalar? And why does the target transcription rate for a module depend on the gene?

Methods, Equations (14-16), can these be written in a more compact and readable form by defining some terms that occur several times in the equations?

Methods, Equations (18-20), does that imply that transcription always start at 0? Since the data we sample usually covers some arbitrary window of a cellular process, this assumption won't always be satisfied in practice. I know that this is an assumption that many competitions velocity models make as well; nevertheless, it would be nice to point this out here for future method development to improve.

Methods, Equation (24). I find the terms "biological values" and "measured values" confusing. I would rather prefer using more standard terminology, like "measured" and "expected". Also, I think it could help readability to not use the superscripts "B" and "M", but maybe to decorate measured values with a symbol, like a $\tilde{}$ or a $\hat{}$.

Implementation

I would strongly advice against storing jupyter notebooks in the main code repository. It inflates the repo unnecessary and increases the download time. I recommend creating a separate repo for notebooks, and re-creating the code repo to remove jupyter notebooks from the history as well.

Please add unit tests and an API reference to make the package maintainable into the future, and easily useable by others.

Installation test:

Installation from github as described in the authors repo works. I created a new conda environment running python 3.9 and installed using `pip install -e .`. However, when I try to import cell2fate in a jupyter notebook via `import cell2fate`, I get

...

RuntimeError: jaxlib version 0.4.18 is newer than and incompatible with jax version 0.4.10. Please update your jax and/or jaxlib packages.

...

I tried resolving this with `pip install --upgrade jax`, which worked.

Reviewer #2:

Remarks to the Author:

The Authors present cell2fate, a novel method to perform RNA velocity analyses.

Overall, I have a very positive opinion about this work: the article is well structured and clearly presented, and the method proposed is interesting, original and well described.

The Authors convincingly validate their approach in various real datasets, comparing results to several alternative RNA velocity tools.

I think that this work represents a significant contribution to the field, and would fit well in Nature Methods.

I have several comments, but (despite the long list) all issues should be relatively easy to address (although some will take time!).

Despite the extra work, I hope the Authors can find my comments useful.

If you re-submit a revised manuscript, please highlight changes in the text so that we can easily identify them (thanks!).

Simone Tiberi, The University of Bologna

Now, the rough part.

Major comments:

1) My main comment is about the presentation of the method.

While this is a methodological article, the method is only briefly and vaguely described in "The cell2fate model" Section. From the main text, I only partially understood the assumptions, formulations and inference of the approach.

I don't think it's ok to keep all methodological aspects in "Methods".

I believe that a longer description of the method should be added in the main paper; particularly focusing on:

- the key rationale of the approach,
- some key formulations such as the ODEs and their linearisation,
- how parameters are inferred, and
- the intuition behind how evidence is shared across genes, cells and modules (mentioned at the bottom of page 2).

Is the sharing of evidence done via an Empirical Bayes approach (i.e., prior estimated from data itself)?

2) In the Methods Section, some details are missing; in particular the full likelihood and full posterior of the model are not explicitly written.

Also, in 1.7, the "stochastic variational inference" approach, which used to infer the model parameters, is only mentioned; instead, I think its intuition of this approach and how it is applied in cell2fate should be briefly explained.

3) While Supplementary Figure 2 reports some runtimes (as far as I understand, only relative to the maximum number of modules calculation), a comprehensive computational benchmark is missing.

It would be appropriate to assess the full runtime and memory usage of the entire cell2fate pipeline (start-to-end), and of all competitors on at least 1 real dataset (possibly more).

4) I found the analysis of the "steady state system" very interesting (as in Fig 2.F).

I would appreciate a Supplementary Figure showing the results from all the other RNA velocity methods as well.

This would be useful to see if/what methods provide spurious RNA velocity directions, even in a steady state model.

Minor comments:

1) At page 2, when explaining the model parameters ("the dynamics of each module ..."), all subscripts should be introduced (i.e., "cell c ", "module m " and "gene g ").

2) The Authors use an ON/OFF switch model: please specify that transcription only happens in the ON state (some switch models allow for transcription to happen in both states).

3) Page 2: "as well as corresponding rates λ ..." -> "as well as corresponding switch rates λ ...". (Rates could refer to switch, transcription or degradation rates).

4) All genes have the same ON/OFF switch rates; I think most (all?) RNA velocity methods share this assumption. Nonetheless, it's still worth briefly acknowledging that this is a simplification.

5) Similarly to the comment above, s and u reads are estimated and carry a degree of uncertainty. While (as far as I am aware), this issue is not accounted for by any RNA velocity method, I think it should be mentioned.

6) The Figures and text often refer to “counts”.

What is your definition of “counts” here: spliced only, or all counts (spliced + unspliced)?

Unless I missed it (quite possible), please clarify this explicitly both in the text, and in the software documentation/tutorial.

7) You use the “cross boundary direction correctness”; I think it is worth briefly explaining what this metric is/measures.

8) Link to the cell2fate software is missing; probably worth adding it in “Code availability”.

9) I found a few typos:

- Page 3 (Fig 1 caption) “allows to infer” -> “allows users to infer” or “allows inferring”

- Page 4 (title): “estimation of complex of transcription” -> “estimation of complex transcription”

- Page 7: “we expected that ... can provide ...” -> “we expect that ... can provide ...” or “we expected that ... could provide ...”

- Page 15 “A.A. conceived of the ...” -> “A.A. conceived the ...”

- Page 15 (A.A. is repeated twice) “... co.supervised A.A., A.A., ...” -> “... co.supervised A.A., ...”

- Methods -> please double check the punctuation after the formulas: sometimes it’s missing, written on the next line, or written twice (after formula and on next line).

- Methods, formula (4) “a_g” -> “\alpha_g”

- Methods, line (186) “is denoted a module” -> “is denoted by a module”.

Suggestions (ignore if you disagree):

1) If I am not mistaken, the average time the gene is active (in the ON state) is given by $\lambda_{ON} / (\lambda_{ON} + \lambda_{OFF})$.

It seems like a useful interpretation point to mention.

2) Sub-Figures are labelled with upper case letters in the images (e.g., A), and lower case in the text (e.g., a).

I’d suggest always using the same approach for consistency.

3) In Figure 2.E (and in the software), the plot title says “Time Uncertainty”.

While “uncertainty” is an easy word to digest for biologists, I think it would be more precise if the software used what it actually plots: CV.

4) The Git repositories with the scripts can be linked to zenodo, and associated to a DOI.

Reviewer #3:

Remarks to the Author:

In this manuscript, Alexander et al. proposes cell2fate as a new RNA velocity model by considering time-varying unspliced mRNA transcription rates. The central assumption of cell2fate is that the gene transcription rates can be decomposed into the linear combination of gene-module specific dynamics, which are activated sequentially in various “time window” during development process. The parameter estimation was performed using Bayesian approach through pyro, and the method was compared with existing methods on real datasets. The method has also been applied to spatial transcriptomics data by inferring on non-spatial scRNA-seq counterparts first, and then project the results to spatial locations use cell2location method.

Generalizing the RNA velocity model to more complex scenarios, e.g. to incorporate cell-specific gene regulation have been developed in several existing literatures. Well-known methods include CellDancer (PMC10545816), Dynamo (PMC9332140), and VeloVAE. The key contribution of cell2fate is to incorporate the gene module concept into RNA velocity analysis, and use its sequential activation dynamics to reveal cell state transition processes. This idea is novel and interesting. However, this feature of cell2fate seems not be fully explored in current manuscript (see comments below). There are also several concerns suggested to be addressed by authors before recommending its publication to Nature Methods.

Major Concern 1: Rationale of assumption and scope of application.

The central assumption of cell2fate is that the genes are activated sequentially in modules. While the assumption seems appropriate for datasets shown in the manuscript, there are competing or alternative assumptions such as CellDancer’s relay model or the bursty model of TopicVelo (PMC10312759). It would be insightful to 1) further compare with these competing alternative models 2) develop diagnostic statistics to evaluate the goodness of fit to real dataset. In addition, the claim that cell2fate is a “biophysically accurate model” and “mathematically sound” seems too strong, since all RNA velocity models, including cell2fate are built upon different assumptions and their relevance to real biological scenario might be case-dependent. Also, the word “model-based” in current manuscript title appears vague and confusing, given that all RNA velocity methods are somehow model-based indeed.

In addition, the absolute values of gene-specific splicing/degradation rates (beta and gamma) of each gene is not possible to solely determined from scRNA-seq data (PMC9332140)– this is a well-known problem about “invariant of scales” in RNA velocity models (pubmed: PMC9550177). Would this bring challenge for the foundation of cell2fate? Could author explain more that how cell2fate overcomes such limitation and obtain the gene-shared unit time?

Major Concern 2: Systematic benchmarking with ground-truth.

The current manuscript has compared the algorithm with existing methods only on real datasets, using CDBir metric. This metric seems indirect given that the ground truth velocities in the datasets are not known quantitatively in real dataset. Could the author include more quantitative benchmarking on simulated datasets where ground truth velocities? In addition, what about the scalability of cell2fate with the increasing size of dataset?

Major Concern 3: Sensitivity analysis and robustness test.

In the proposed algorithm, there are numerous heuristic choices of parameters especially regarding the prior distribution. Another important hyper-parameter is the max number of modules and is selected empirically by leiden cluster number times 1.15. However, their influence on performance and algorithm robustness has not been studied in current manuscript. Extensive study on parameter sensitivity and cell2fate robustness seems very necessary for the broader application of the method.

Major Concern 4: New biological insights.

Considering gene module dynamics as a key novelty of cell2fate, the downstream analysis on its biological implication and insights seems not adequate in current manuscript. For instance, could module dynamics imply about the “driver genes” in the transition? Are there specific genes that have significant components in multiple modules, and what is the biological interpretation? In other words, if the gene expression dynamics are all solely dominant by one specific module, would the cell2fate simply reduce to scVelo model, by assuming a single on-off switch? Explorations like this could greatly enhance cell2fate’s interests to general readers and validate its rationale of sequential module activation assumption.

Major Concern 5: Detailed documentation and API.

The provided github repo includes the annotated jupyter notebook for tutorial along with codes to reproduce results. In the meantime, the major improvements are needed to include detailed documentation about functions and APIs in cell2fate package.

Major Concern 6: The Methods file needs significant revision to meet Nature Methods standard

- 1) The Methods file is not well formatted, along with quite some typos. Overall, it appears overly technical and lengthy for general readers of Nature Methods – the authors may consider split some part into supplementary notes. Format-wise, many sentences involving equations are starting with the comma (,), e.g. line 396. There’s also issue with paired double quotation marks (“”), e.g. line 382. Other typos include the notation error on line 465 and reference error line 470.
- 2) Some claims seem highly speculative to appear in Methods. For example, the comment about stochastic modelling (line 250-253) lacks convincing validation.

Minor :

1. Relevant to Major point 2, does the proposed method filter “velocity genes” based on “goodness of fit” (similar to scVelo) to plot streamlines? Does different highly-variable gene selection affects the results?
2. The gene names in Supp. Fig. 23 are too small to be recognized.
3. Have the gene expression values in Fig. 3B and Fig. been smoothed?

Version 1:

Decision Letter:

Our ref: NMETH-A53382A

5th Sep 2024

Dear Dr. Bayraktar,

Thank you for submitting your revised manuscript "Inference of RNA velocity modules improves cell fate prediction" (NMETH-A53382A). It has now been seen by the original referees and their comments are below. The reviewers find that the paper has improved in revision, and therefore we'll be happy in principle to publish it in Nature Methods, pending minor revisions to satisfy the referees' final requests and to comply with our editorial and formatting guidelines.

When submitting the revised paper, please also provide a point-by-point response letter.

TRANSPARENT PEER REVIEW

Nature Methods offers a transparent peer review option for new original research manuscripts submitted from 17th February 2021. We encourage increased transparency in peer review by publishing the reviewer comments, author rebuttal letters and editorial decision letters if the authors agree. Such peer review material is made available as a supplementary peer review file. **Please state in the cover letter ‘I wish to participate in transparent peer review’ if you want to opt in, or ‘I do not wish to**

participate in transparent peer review' if you don't. Failure to state your preference will result in delays in accepting your manuscript for publication.

ORCID

Sincerely,

Lin Tang, PhD
Senior Editor
Nature Methods

Reviewer #1 (Remarks to the Author):

The reviewers greatly improved their manuscript and satisfied almost all of my previous comments. The only (small) point left from my side is their scalability evaluation (both for time and memory) - I think this can be visualized in a much more effective way than in the current tables, which make it hard to interpret scalability in the number of cells and genes (consider e.g. scatter plots, bar charts - anything that makes it possible to evaluate scalability visually). The authors claim linear scalability, it would be nice to also demonstrate this practically (the authors have the results already).

Also, methods that make use of parallelization (like cellDancer) should be run in parallel - it seems unfair to run GPU-optimized methods on GPU, but to not run CPU-optimized methods in parallel, if they support parallelization.

In summary, I'm happy with the revised version and I recommend publication in Nature Methods.

Reviewer #1 (Remarks on code availability):

I've reviewed the code in the first iteration and the authors have resolved my comments.

Reviewer #2 (Remarks to the Author):

The Authors have suitably implemented (or replied to) all my comments.

I'd like to thank the Authors for the time and effort they invested to implement the suggested corrections and clarifications. In the past, I've worked with ON/OFF 2-state transcription models, and I had partially misunderstood some aspects of cell2fate (interpreting it based on my previous knowledge). The added explanations make the context clearer.

I only have an additional minor request regarding the mathematical rigour of the formulations in the Supplementary (and partially in the main text).

- distributions are usually expressed with tilda (\sim) instead of equals ($=$).
E.g., $X \sim \text{NegativeBinomial}(\dots)$ (formula (8)).

- densities are usually denoted by $f(\dots)$.
e.g. (at Supp formula (37)) $P(X = x; \theta) = f_{\text{NB}}(x; \theta)$, where $f_{\text{NB}}(\cdot; \theta)$ indicates the density function of a Neg Bin distribution with mean ... and variance ...

- the likelihood in Supp formula (37) is only written with respect to X_{cgj} .
That is the density of X_{cgj} though, not of the entire dataset.
The full likelihood is likely expressed as the (possibly treble) product of that density over indices c , g and j .

Furthermore, theta lacks subscripts; since theta varies depending on subscripts cgj, I think it'd be written as theta_cgj for X_cgj. While the full theta vector is likely the aggregation of such theta_cgj elements (please define it clearly).

Typo:

- Page 4: which vastly reducing -> which vastly reduces

P.s., "Ignore" is the more fun reply I ever received as a reviewer.

Fair enough: after replying to 15 comments, it seems comprehensible.

Nonetheless, I still think that in the caption of Figure 2 the word "uncertainty" should be clarified (uncertainty is a vague term that can be measured in multiple ways).

Simone Tiberi,
The University of Bologna

Reviewer #3 (Remarks to the Author):

In the revised manuscript, the authors have done a good job of addressing the previous concerns, especially on improving the robustness of method. I have no further questions and recommend the acceptance of this manuscript.

Reviewer #3 (Remarks on code availability):

The cell2fate package is a usable resource for the community with enough instructions for installing and running the applications. I have no problem to run the test codes on GPU.

Version 2:

Decision Letter:

23rd Jan 2025

Dear Dr Bayraktar,

Thank you very much for sending us the updated version of the main text and Supplementary Information files. I am pleased to inform you that your Article, "Cell2fate infers RNA velocity modules to improve cell fate prediction", has now been accepted for publication in Nature Methods. The received and accepted dates will be 2nd Aug 2023 and 23rd Jan 2025. This note is intended to let you know what to expect from us over the next month or so, and to let you know where to address any further questions.

Over the next few weeks, your paper will be copyedited to ensure that it conforms to Nature Methods style. Once your paper is typeset, you will receive an email with a link to choose the appropriate publishing options for your paper and our Author Services team will be in touch regarding any additional information that may be required. It is extremely important that you let us know now whether you will be difficult to contact over the next month. If this is the case, we ask that you send us the contact information (email, phone and fax) of someone who will be able to check the proofs and deal with any last-minute problems.

You may wish to make your media relations office aware of your accepted publication, in case they consider it appropriate to organize some internal or external publicity. Once your paper has been scheduled you will receive an email confirming the publication details. This is normally 3-4 working days in advance of publication. If you need additional notice of the date and time of publication, please let the production team know when you receive the proof of your article to ensure there is sufficient time to coordinate. Further information on our embargo policies can be found here:

<https://www.nature.com/authors/policies/embargo.html>

If you are active on Twitter/X, please e-mail me your and your coauthors' handles so that we may tag you when the paper is published.

Please feel free to contact me if you have questions about any of these points. Thank you very much for publishing your paper at Nature Methods!

Best regards,

Lin Tang, PhD
Senior Editor
Nature Methods

** Visit the Springer Nature Editorial and Publishing website at http://editorial-jobs.springernature.com?utm_source=ejP_NMeth_email&utm_medium=ejP_NMeth_email&utm_campaign=ejp_Nmeth for more information about our career opportunities. If you have any questions please click [here](mailto:editorial.publishing.jobs@springernature.com).

Open Access This Peer Review File is licensed under a Creative Commons Attribution 4.0 International License, which permits use, sharing, adaptation, distribution and reproduction in any medium or format, as long as you give appropriate credit to the original author(s) and the source, provide a link to the Creative Commons license, and indicate if changes were made. In cases where reviewers are anonymous, credit should be given to 'Anonymous Referee' and the source. The images or other third party material in this Peer Review File are included in the article's Creative Commons license, unless indicated otherwise in a credit line to the material. If material is not included in the article's Creative Commons license and your intended use is not permitted by statutory regulation or exceeds the permitted use, you will need to obtain permission directly from the copyright holder.

Response to Reviewers' Comments:

Please find our responses to the three reviewer's comments below. On our manuscript, we have marked the edited text in red.

Reviewer #1:

Remarks to the Author:

Summary

The authors present cell2fate, a Bayesian model to estimate RNA velocity in a modular fashion. Their main innovation is to parametrize the transcription rate as a sum over different modules, which induces a modular structure for spliced and unspliced counts as well. With this formulation, the ODEs for each module can be solved analytically, and model parameters can be fit using stochastic variational inference. Their formulation of RNA velocity offers the following advantages:

- Operating on the level of gene modules makes a lot of sense. We know that genes act in modules for many biological processes. This allows for more flexible dynamical patterns of the transcription rate, while remaining computationally tractable. Framing the problem this way is both biologically as well as numerically a useful approximation.
- Operating directly on the level of raw mRNA counts, avoiding imputation.
- Explicitly accounting for technical effects, such as batch effects, ambient mRNA, and detection sensitivity of spliced/unspliced mRNA.
- Fitting in a fully Bayesian fashion, which allows for the inclusion of useful information via priors (e.g. sparsity, effective number of active modules, etc.). Also, this allows the authors to estimate velocity and parameter uncertainty.

Their method is a useful addition to the suite of RNA velocity models, and shows promising performance in a number of applications and benchmarks. The idea of combining velocity inference with a membership-like model is innovative and offers several avenues for future improvements, as the authors lie out in the discussion. I think this work will be of broad interest to biologists and bioinformaticians working with single-cell data from various dynamical processes.

We appreciate the reviewers enthusiasm for cell2fate and thank them for recognising the innovation and potential for future developments.

I have a number of concerns relating to **the metrics used for benchmarking, visual inspection of velocity streams**, manual definition of interesting trajectories, method scalability, and the choice of datasets (see my major comments below). Most importantly, it would be great to see cell2fate's performance in a more challenging real-data setting, like regeneration, reprogramming or cancer, where simpler approaches fail.

Thank you for the detailed suggestions for extending our benchmarks. In response to the individual points raised, we have substantially extended the assessment of cell2fate. Specifically, we now consider five complementary metrics and approaches to benchmark cell2fate and (where appropriate) alternative velocity methods.

1. **CBDir assessment of velocity streams:** This is the primary metric considered in our previous revision. While we agree that this metric is far from perfect, it has several appealing properties motivating its use. Most importantly, this quantitative metric can be consistently applied to all datasets based on standard velocity output that are produced by any of the considered methods. Additionally, it is straight-forward to define a consistent ground-truth because it only assesses transitions between clusters.

The core limitation of CBDir is that it is based on a 2 dimensional UMAP representation, thus inheriting all the possible issues of UMP (see for example DOI: [10.1371/journal.pcbi.1011288](https://doi.org/10.1371/journal.pcbi.1011288)). Nevertheless, a UMAP representation remains the canonical visualisation of velocity estimates, and hence it remains relevant. CBDir however does not allow for assessing transitions across finer subclusters and it remains an indirect assessment of velocity, as it relies on cell transition probabilities and does not assess the accuracy of estimated dynamical rates (for which there is usually no ground truth available).

To mitigate the limitations of the CBDir benchmark, we now consider additional complementary metrics and benchmarking strategies:

2. **Consistency of inferred time with known cell-state transitions:** As an alternative metric to CBDir, we have assessed the consistency of cell-specific time estimates with known cell state transitions. Briefly, we considered this metric to assess cell2fate, as well as the pyroVelocity_model2 and UniTVelo_unified - the top performing methods in the CBDir benchmark, which also yield cell-specific rather than gene-specific time estimates. We then assess the difference in inferred time between clusters in the light of known cell state transitions (c.f. Supp. table 8).; see response to comment R1.1
3. **Consistency of inferred time with developmental time point of sample collection:** two out of the five benchmark datasets comprise cells sampled at multiple (known) developmental stages, which can serve as additional ground truth to assess inferred orderings. Analogous to (2), we assessed the estimated cell-specific orderings relative to the developmental times of coarse cell clusters; see response to comment R1.1.
4. **Combination of velocity estimates with cellrank:** As mentioned by the reviewer, Cellrank can be applied downstream of cell2fate and other velocity methods, which provides for an alternative approach to assess velocity streams. We have benchmarked the combination of cellrank with all of the considered velocity methods, thereby confirming the key claims concerning specific transitions that were previously based on visual inspection of UMAP plots; see our response to comments R1.2 and R1.3.
5. **Memory and computational complexity:** We have compared alternative velocity methods in terms of compute and memory requirements; see response to comment R1.4.

6. **Simulation study:** In response to reviewer comment R3.2, we fit cell2fate to simulated data to confirm the method can identify ground-truth dynamical rates (and not only correct time and cell transition probabilities).

As part of the revisions, we have also shown how the posterior samples from cell2fate allow for estimating a confidence score for cell state transitions between clusters. We discuss this metric in more detail in response to comment R3.1.

Major comments

R1.1) Fig. 2, benchmarking of velocity methods. I appreciate the authors' efforts to compare 10 different velocity models across 5 different datasets, but I'm not fully satisfied with their metrics. The choice CbDir seems appropriate, yet I would encourage them to include another quantitative measure to evaluate their trajectories. For example, the authors could work more with their cell-specific latent time, and check whether it correctly increases with increasing maturity. This could be done in at least two ways: (1) for datasets that contain experimental time points, they could check whether their latent time increases from early to late time points, (2), for single-snapshot datasets, they could define a very coarse ground-truth ordering of cell populations/types (which should be easy e.g. in hematopoiesis), and compare the temporal ordering indicated by their latent time with ground truth, using e.g. spearman rank correlation between the ground truth ordering and the mean latent time per cell type. Their current panel E goes in this direction, but does not include any quantitative evaluation.

We thank the reviewer for both of these suggestions, which we have taken into consideration when revising our manuscript. Since not all RNA velocity methods compute (our output) an explicit cell-specific time estimates, we have limited this analysis to cell2fate as well as pyroVelocity_model2 and UniTVelo_unified (c.f. Supp. table 8). However, notably the results and the relative performance of these methods across the five benchmark datasets is almost entirely consistent with the results based on the low-dimensional CbDir metric, indicating that the CbDir metric is adequate for quantifying the directionality of transitions in our 5 benchmarking datasets.

Regarding your suggestion to exploit known developmental time points, we have applied this strategy to two datasets that recorded developmental time for individual samples subjected to scRNAseq profiling - the Erythroid maturation dataset, which contains cells from 7 mouse developmental time points (between days 7 and 9) and the Dentate Gyrus dataset, which contains cells from two days (day 12 and day 35). For both datasets, we employed granular Leiden clustering to identify transcriptionally similar cell states and calculated average developmental time per clusters, which we compared to average cell-specific time estimates obtained from velocity models.

For cell2fate, the average developmental and estimated time were in striking agreement for both datasets ($r=0.88$, pearson R) (**Fig R1.1, new Supp. Fig 20 and 21**); the other two methods achieved a positive correlation only on the Erythroid maturation trajectory. In the Erythroid maturation data, the cluster that deviated the most from our inferred process time was a progenitor cluster that contained many cells from the latest developmental time point,

which is expected since self-renewing progenitor cells do not necessarily progress through the differentiation process during development. In the Dentate Gyrus dataset the average developmental time of mature granule neuron clusters increased in line with cell2fate's inferred process time, which provides further evidence for the inferred late maturation trajectory that was identified by cell2fate but not detected by other methods (**Fig 2B**). These results are presented as supplementary figures 20 and 21, and we discuss these results in the revised manuscript text (page 9, paragraph 2).

Figure R1.1: Comparison of developmental age of mouse samples to time estimates in the Dentate Gyrus dataset **A:** Developmental age of each cell (top) and Leiden clustering (bottom). **B to E:** Estimated time for each cell (top) and average estimated time vs. average developmental age of cells in Leiden cluster (bottom)

Regarding the second suggestion, we have computed the differences in median cell-specific time estimates across clusters in the five datasets to validate the consistency of cell-specific time estimates with respect to known cell state transitions. Reassuringly, this analysis applied to the cell2fate results yielded positive time differences for 18 of 19 previously described cell transitions. The only exception is 'HSC_1' to 'HSC_2' cluster transition in the Human Bone Marrow dataset (see table below), for which cell2fate infers the wrong directions, which is consistent with the CBDir metric. The pyroVelocity_model2 resulted in 14 transitions with positive time differences, which again were consistent with the CBDir metric. The UniTVelo model gave slightly different results for the two metrics, yielding 10 transitions with positive time difference, but 14 with positive CBDir metric. We present these results in supplementary table 8, shown as table R1.1 below and mention it in the adapted manuscript text. We discuss the confidence score in the table in more detail in our response to R3.3.

Dataset	Transition	cell2fate		cell2fate pyroVelocity_model2	pyroVelocity_model2	UniTVelo_unified		
		Time Difference	CBDir			Time Difference	CBDir	
Pancreas_with_cc	('Ngn3 high EP', 'Pre-endocrine')	18.67	0.84	0.99	14.221233	0.75700743	0.3275501	0.738413507
Pancreas_with_cc	('Pre-endocrine', 'Alpha')	25.10	0.79	0.97	26.139313	0.727597414	0.09182978	0.768862605
Pancreas_with_cc	('Pre-endocrine', 'Beta')	40.30	0.74	0.88	26.800117	0.802793827	0.13858652	0.828286199
Pancreas_with_cc	('Pre-endocrine', 'Delta')	14.64	0.25	0.97	35.815857	0.525459787	-0.07874572	0.426446807
Pancreas_with_cc	('Pre-endocrine', 'Epsilon')	16.96	0.34	0.81	8.995232	0.065836651	-0.008022428	-0.014511983
DentateGyrus	('nIPC', 'Neuroblast')	5.63	0.27	0	6.1603165	0.912278111	-0.17385006	-0.897451104
DentateGyrus	('Neuroblast', 'Granule immature')	27.82	0.65	0.93	-5.843239	-0.157438249	-0.37807927	-0.562275853
DentateGyrus	('Granule immature', 'Granule mature')	10.23	0.60	0.4	-1.3436089	0.112992168	-0.06260124	0.183849973
DentateGyrus	('Radial Glia-like', 'Astrocytes')	3.37	0.53	0.26	-1.1435738	-0.746837717	-0.053927362	0.820815285
DentateGyrus	('OPC', 'OL')	13.17	0.64	0.98	6.54385	0.893879764	0.17046958	0.809825533
MouseBoneMarrow	('dividing', 'progenitors')	8.27	0.59	0.96	11.932587	0.89050394	-0.22601354	-0.863546662
MouseBoneMarrow	('progenitors', 'activating')	11.33	0.75	0.91	-3.5885544	-0.562520382	-0.13892986	-0.783345283
MouseErythroid	('Blood progenitors 1', 'Blood progenitors 2')	7.54	0.45	0.92	1.9548702	0.909141695	0.0768282	0.926103739
MouseErythroid	('Blood progenitors 2', 'Erythroid1')	8.92	0.82	0.93	6.765785	0.806573288	0.31297496	0.762791743
MouseErythroid	('Erythroid1', 'Erythroid2')	7.53	0.86	0.94	15.823822	0.84265669	0.21306533	0.825326534
MouseErythroid	('Erythroid2', 'Erythroid3')	8.78	0.88	0.95	37.07811	0.831446314	0.14975423	0.829991239
HumanBoneMarrow	('HSC_1', 'Ery_1')	35.06	0.11	1	-0.15167236	-0.108933346	0.32684237	0.882813249
HumanBoneMarrow	('HSC_1', 'HSC_2')	-13.69	-0.32	0.03	0.42607117	0.342610081	-0.17935735	0.671795387
HumanBoneMarrow	('Ery_1', 'Ery_2')	22.10	0.53	0.92	2.1642113	-0.017749694	0.097133994	0.875854767
Correct Transitions		18	18	n.a.	14	14	10	14

Table R1.1: Median time difference between cells in clusters involved in cell state transitions. Positive differences indicate consistency with prior knowledge. The score indicates what percentage of posterior samples in the second cluster have higher time values than the 90th percentile of cells in the first cluster. CBDir values are included for comparison.

R1.2) Projected velocity streams over-interpretation: I would also urge the authors to rely less on projected velocity streams in Panels B and C of Fig. 2, as these can be very misleading, as noted in several previous publications. **The same holds for a few other panels throughout the manuscript (e.g. SFig. 22 and others). I think it would be better to evaluate the resulting trajectories quantitatively in high dimensions, using tools like CellRank or dynamo.** Especially for small populations like the OPCs, I think projected streamplot on a UMAP convey very little information and are prone to over-interpretation. For example, L215, "(...) while radial glial-like progenitor cells are largely committed to astrocytes", if that statement has been derived from eye-balling the UMAP in Fig. 3A, then I would strongly advice for using a more quantitative approach of estimating fate probabilities.

We agree with the reviewer that UMAP velocity streams can be misleading due to the intrinsic limitations of UMAP-based projections. Having said this, UMAP-based interpretations of velocity streams remain widely adopted in the field, justifying the use of this metric. As a complementary metric, we have applied CellRank to quantitatively evaluate trajectories from Cell2fate and alternative methods. We focused this analysis on two datasets from Dentate Gyrus development and Erythroid maturation that our original manuscript examined in detail regarding their cellular transitions and transcriptional kinetics (Fig 2B,C).

Overall, the analysis based on CellRank largely confirmed the cellular trajectories implied by the UMAP projections shown in figures 2B, C, confirming the observation that alternative methods missed several key cell state transitions (Sup Figs 14-17, Fig R1.2 and R1.3 below).

Specifically, the analysis on the Dentate Gyrus dataset yielded three insights. Firstly, only cellRank with cell2fate velocity inputs identifies both Astrocytes, Oligodendrocytes and Granule Mature cells as macrostates. CellRank with velocity inputs from other methods missed the Granule Mature macrostate and often also the Astrocyte state. Secondly, cellrank with cell2fate velocity input assigned high fate probabilities to the correct progenitor cells of

each of the three terminal states, such as high probability for the Granule mature state for nIPCs and lower for Radial Glia-like cells, which are instead predominantly assigned to the Astrocyte state. We also include a scatter plot with probabilities below (Figure R1.3) and in supplementary figure 24A. Thirdly, cellrank with cell2fate velocity input also assigned high probabilities for the Granule mature terminal state to the OPC cells, rather than high probabilities for the OL terminal state alone, thus revealing a limitation of the cell2fate velocity estimates. In contrast to these results from CellRank, the OPC to OL transition does have the correct median time difference as previously shown in our response to R1.1. Hence it is possible that the OPC to OL velocity flow is overall correct in its direction, but very weak (maybe because OPCs also contain dynamics from the cell-cycle). Nonetheless, because of these conflicting results we have removed the claim from the main text and figure 2B that cell2fate can identify correct velocity flows in OPCs.

For the Erythroid maturation dataset, we also ran CellRank with default parameters and computed fate probabilities both towards the progenitor 1 state and the Erythroid 3 state. This confirmed that cell2fate velocity suggests an overall trajectory towards the Erythroid 3 state, unlike some other methods, such as pyro-velocity_model1, which showed significant fate probabilities for the progenitor 1 state across all cells, when combined with CellRank.

A**B****C**
Figure R1.2: CellRank Macrostates and fate probabilities for RNA velocity input from three methods. Only cell2fate identifies mature granule neurons as a macrostate and can thus calculate fate probabilities for this state.

Figure R1.3: Astrocyte and Granule neuron fate probabilities: Radial_Glia-like cells are mostly committed to Astrocytes and nIPCs to Granule neurons.

R1.3) Defining trajectories: In Fig. 3, it appears as if the authors had to manually subset their data to characterise the granule maturation trajectory. The same appears to be the case in Fig. 4 for the excitatory neuronal lineage. Could this be automated by e.g. using a tool like CellRank or dynamo to compute fate probabilities, and then select those cells which are likely to become granule neurons (as opposed to Astrocytes)? Computing fate probabilities to differentiate between different trajectories in an unbiased fashion is an important use case of RNA velocity.

Yes this indeed possible and we have included such an analysis in the revised manuscript (c.f. Response to comment R1.2). The analysis based on cellRank confirms the general conclusion that Radial glia-like cells are mostly committed to Astrocytes, unlike nIPCs. Our previous conclusion was drawn from the observation that Radial-Glia cells are mostly

assigned high time values, which are close to Astrocytes, rather than low time values, which are close to nIPCs. Below we plot cell2fate inferred time vs. cellrank ratios of Astrocyte vs Granule mature fate probabilities for all Radial Glia cells. We included this plot as a supplementary figure 23B and mention it in the manuscript section 3.

B

Figure R1.4: Comparison of cell2fate time estimates and cellRank (on cell2fate velocity input) ratio of Astrocyte to Granule Neuron fate probabilities for Radial Glia cells: Cells with high probability for the Astrocyte fate tend to have a higher inferred time by cell2fate. Overall, almost all Radial Glia cells are committed to Astrocytes.

R1.4) Method scalability, see Methods, Fig. 2. I expect this model to be computationally more expensive compared to scVelo's dynamical model, especially when using parallelization for scVelo. On a dataset of 10k cells across 3k genes, on a modern MacBook with 8 cores, I expect scVelo's dynamical model to finish in around 7-10 minutes. It would be good to have some sort of comparison here (in terms of runtime and memory consumption) with a baseline model to illustrate the trade off between the additional complexity of cell2fate, and increased computational cost. Also, it would be good to investigate how compute time scales with the number of cells, modules and genes, either theoretically or through experiments. I suppose this model is currently geared towards small-scale single cell datasets containing less than 50k cells (the largest dataset in this study is below 10k cells). It would be good to state this as a limitation in the discussion, if that's the case, and to discuss how future improvements could make the model more scalable and applicable to atlas-sized datasets. In principle, batch effect correction is a very valuable component to achieve this goal, so cell2fate could be well positioned if it achieves the necessary scalability.

We have now included a comprehensive run time and memory benchmark in supplementary tables 6 and 7, which are also included below.

First, we note that the computational complexity of cell2fate scales linearly in the number of cells and genes, as do other methods, because the number of parameters that need to be inferred scales linearly with those variables. The memory requirements decrease linearly

with the batch size used during training, whereas run time increases with decreasing batchsize (cell2fate currently uses a fixed batch size of 1,000 cells). Second, as shown in an empirical comparison of the computational and memory requirements, the cost of running cell2fate is in line with alternative methods (Table R1.2 and R1.3).

Critically, in practice, scalability is not limiting biological discovery, since the model can be applied in its current version to atlas sized datasets of >1 million cells using a single standard GPU in less than 48 hours. We base this claim on the results on the 10**4 cells Erythroid maturation dataset, where cell2fate requires 1,470 seconds (0.4 hours). This implies it could be run on a 10**5 cell dataset in 4 hours and 10**6 cells in approximately 40 hours.

In the future, if datasets grow even beyond current resources, we note that the efficiency of cell2fate could be further improved if needed. Promising avenues for optimizations include replacing the time-consuming pre-processing step of clustering the data to determine an appropriate number of modules. This step can be skipped if the data has already been clustered in a previous analysis. Counting the number of PCs that explain a certain threshold of variation in the data could be employed as a fast heuristics to determine an appropriate number of modules. Secondly, sampling 100 samples from the posterior distribution is time consuming for datasets with large numbers of cells. Instead, it is also possible to only extract the median, 25th and 75th quantile of the posterior of each parameter. Thirdly, data-driven criteria for early stopping of training could be employed, for example requiring a certain threshold value for a minimal loss improvement. Finally, amortisation can provide an opportunity for faster convergence on large datasets by not optimising each cell-specific parameter separately, but by instead learning a function that predicts the parameters of each cell, based on its expression. We have experimented with such speedups, however given that cell2fate is already practically applicable to large datasets, we feel that such extensions are an area of future work. We have included this in the discussion section.

	DentateGyrus	HumanBoneMarrow	HumanDevelopingBrain	MouseBoneMarrow	MouseErythroid	MousePancreas
CellDancer	13681.3	23736.6	36760.6	4171.3	38191.0	17220.5
cell2fate	561.0	557.8	590.1	340.8	1469.8	692.6
scVelo	577.8	935.9	1586.1	227.3	1073.0	896.8
DeepVelo	908.2	1827.1	3090.0	280.9	3139.3	1163.8
pyroVelocity_model1	104.1	323.5	801.6	62.7	764.3	120.2
VeloVAE	283.2	325.7	349.6	219.4	612.3	211.3
pyroVelocity_model2	175.2	277.1	484.7	93.1	1012.3	517.6
scVelo_stochastic	3.4	6.4	19.7	1.3	61.3	6.2
VeloVI	121.9	252.1	283.1	99.3	265.3	232.8

Table R1.2: Computation time on a Tesla V100-SXM2 32GB GPU for *both* data preprocessing, model training and velocity calculation for all methods across all datasets. Results are in seconds. Table is also added as supplementary table 6 and also mentioned in the revised main text. Overall, whereas scVelo was indeed faster than cell2fate, cell2fate performed comparable to other GPU accelerated methods. Specifically, cell2fate completed in 341 seconds on the smallest 2600 cell Mouse Bone Marrow dataset and in 1470 seconds on the largest 9815 cell Mouse Erythroid Maturation dataset.

Method	Memory(MiB)	GPU	n_jobs
CellDancer	1515	No	1
cell2fate	7271	Yes	1
DeepVelo	6397	No	1
pyro-velocity_model1	4757	Yes	1
scvelo_stochastic	1660	No	1
scvelo	988	No	1
pyro-velocity_model2	9253	Yes	1
VeloVAE	1931	Yes	1
veloVI	1249	Yes	1
UniTVelo_unified	1092	No	1
UniTVelo_independent	1112	No	1

Table R1.3: Memory requirements during model training on the largest datasets (Mouse Erythroid). This table is also added as supplementary table 7 and mentioned in the revised main text. Again cell2fate performs comparable to other GPU based methods, for example using less memory than pyro-velocity_model2, but more than pyro_velocity_model1. We also include information about whether a method used the GPU or CPU and clarify that only 1 job was run. Methods that do not leverage the GPU, such as CellDancer, are optimised to run in parallel on multiple CPUs rather than on one GPU. This means that in principle using 100 CPUs could reduce the computation time by a factor of 100 for these methods.

R1.5) Datasets. Currently, cell2fate has been demonstrated on a number of developmental (mouse pancreas, dentate gyrus, and erythroid differentiation and human neurogenesis) and steady-state (mouse and human bone marrow) processes. However, these are arguably processes where simpler methods, like pseudotime inference, work well and provide many biological insights as well. I would be interested to see the performance of cell2fate in a more challenging setting where simpler approaches, like PAGA, slingshot, Palantir, or DPT, fail. Settings like regeneration or cancer are usually more challenging, because we don't know the correct directions a priori, and we cannot just subset to the trajectory we want to study (because that's what we need to discover). These are the settings where RNA velocity is most promising as a concept, and this is where cell2fate should provide new insights that we could not get with competing velocity approaches, or simpler pseudotime approaches.

We feel that the additional value of including such datasets is limited because if the dynamics in a dataset are unknown we cannot validate that cell2fate predictions are correct. One option would be to conduct orthogonal in-vitro experiments on cancer cell lines as a validation, which however is out of scope for this revision and a computational manuscript. Having said this, we note that our benchmark strategy does already cover a number of datasets and biological contexts with a known prior trajectory that entail a particularly

challenging dynamics. These include transcriptional boost patterns in the erythroid maturation dataset (Figure 2C), weak transcription rate changes in mature Granule neurons, low read depth in the Mouse Bone Marrow dataset and slow transcriptional changes in the Human Bone Marrow dataset. We note that in all of these settings, cell2fate performs very well compared to existing alternatives.

We would also like to highlight that in addition to offering accurate predictions, cell2fate also yields a high level of interpretability, which can aid the generation of new insights, for example by dissecting markers of granule neuron differentiation and by mapping RNA velocity modules to spatial transcriptomics data. To illustrate this in the granule cell lineage, we have now also intersected module marker genes with known disease risk genes; see our response to R3.4. In addition, in response to your comment, we now present additional analyses to illustrate the difference between cell2fate's biophysically motivated time estimates and conventional pseudotime estimates. Since pseudotime bases temporal distance estimates on transcriptional distance, we reasoned that the most obvious deviation from true temporal progressions will occur for cells that change only very subtly over time, such as mature granule neurons. In supplementary figure 20, which we include above as figure R1.1, we thus applied standard scanpy diffusion pseudotime to the granule neuron maturation trajectory, finding that unlike cell2fate (red box), pseudotime does not estimate a temporal progression within mature granule neurons. As a result, the sequential gene expression patterns in the final stages of granule neuron maturation cannot be resolved, as we illustrate in our new supplementary figure 22, included as Figure R1.5 below. This shows that cell2fate can be used to uncover new biology even in a widely characterised cell lineage such as granule neurons. We envision that such a physically more realistic estimate of cellular time will become increasingly important in the future for fitting causal models of cellular processes, including gene regulation and cell-cell signalling.

Figure R1.5: Sequential activation of cell2fate module marker genes during final stages of granule neuron maturation ordered by cell2fate time or pseudotime. The expression shown is a moving average of 100 cells of the log₂ transformed and total count normalized UMI.

counts in each cell A: Cells ordered by cell2fate time B: Cells ordered by pseudotime. Only ordering by cell2fate estimated time reveals the sequential gene expression patterns.

Minor comments

R1.6) L46: What do the authors refer to with “unconstrained” when they say “(...) that allows for capturing unconstrained transcriptional dynamics while retaining computational and numerical tractability.” ?

To improve clarity, we replaced “unconstrained” with “realistic”.

R1.7) L50: The following sentence is unclear to me: “Cell2fate employs a linearization to decompose complex differential equations into tractable components that can be solved analytically”. The 2-ODE formulation of RNA velocity, which corresponds to a first-order moment approximation of the Chemical Master Equation (CME), consists of 2 linear, first-order ODEs, which can be solved analytically if we assume constant rates (and one switch for α). The authors should be more explicit at this point about what they mean when they say “complex differential equations.” Why are they complex? One problem is clearly that we don’t measure time explicitly, but that’s a problem with the data, not with the ODE system.

We have significantly revised the method description in the main text to improve clarity; see also response to reviewer comment R2.1. The wording “complex differential equations” is indeed imprecise and we have replaced this statement with a more verbose and precise explanation: “A significant challenge for RNA velocity models is that transcription rates over time follow complex non-linear functions of changes in active transcription factor abundance in the nucleus, yet the integral of the transcription rate function $\alpha_g(t)$ needs to be well-defined to solve the ODEs. As a consequence, existing methods are either based on the assumption of a simplified step-wise functions for $\alpha_g(t)$, or resort to numerical approximations of the ODE solutions.”

R1.8) L63, “Cell2fate builds on a long-standing history of computational methods for RNA velocity (...)” is an overstatement. RNA velocity for single cells has existed since 2018, that’s not really a “long history.”

We note that the concept of RNA velocity actually goes back to mRNA dynamics established in 2011 (Zeisel et al., Molecular Systems Biology, 2011, <https://doi.org/10.1038/msb.2011.62>). However, we agree that terms such as long-standing are subjective and we reworded this sentence.

R1.9) L224, “This analysis identified the successive induction of modules across the early differentiation of radial glia into nIPCs, neuroblasts and immature neurons”, I thought you found radial glial cells to be mostly committed to Astrocytes here?

Please see our answer to your comment R1.3).

R1.10) Fig. 3, I think the legend description for panels B and C is mixed up.

Thank you, this is now fixed.

R1.11) Fig. 3, does the data displayed in Panel B correspond only to the subset of cell highlighted with the red box in A? This is unclear.

Revised text: "*Spliced counts abundance caused by selected modules over time in the Granule neuron lineage, corresponding to cells in the red box in figure A.*"

R1.12) Fig. 3D, I don't really understand this panel. This could be explained better in the Figure and text

We have expanded the explanation in the figure legend.

Additional minor comments on the Methods

R1.13) Methods, Eq (1-3): " (..) these differential equations have been suggested (..)". There is a clear link between Eq (1) and Eqs (2-3). The latter two equations correspond to the first-order moment approximation of the CME corresponding to Eq (1).

Ok we have added this for clarity.

R1.14) Methods L88, describe the problems with parameter fitting here briefly.

Ok we did so in the revised supplementary methods file.

R1.15) Methods, L101, is that also true for the scVelo model? I thought they could somehow be pooled post-hoc to live on the same scale.

Yes, they can be pooled post-hoc in multiple steps, including taking averages across all genes. But the underlying RNA velocity formulation and the model that is fit to the data, treats each gene independently, so we think it is ok to leave the text as it is, since we list the limitations of the RNA velocity "formulation", as stated just before.

R1.16) Methods, L164, which is there a transpose here? Is that not a scalar? And why does the target transcription rate for a module depend on the gene?

The T is not a transpose here, but simply stands for "Target". We replaced the T with a hat to avoid this confusion. The target transcription rate is different for each gene in each module. What is shared across genes is the switch ON/OFF time of each module and the rate at which the target transcription rate is reached.

R1.17) Methods, Equations (14-16), can these be written in a more compact and readable form by defining some terms that occur several times in the equations?

Yes thank you for the idea, we have done this now in the supplementary methods.

R1.18) Methods, Equations (18-20), does that imply that transcription always start at 0? Since the data we sample usually covers some arbitrary window of a cellular process, this assumption won't always be satisfied in practice. I know that this is an assumption that many competitions velocity models make as well; nevertheless, it would be nice to point this out here for future method development to improve.

By considering a sum over multiple modules, cell2fate actually relaxes this assumption that other RNA velocity models have. Specifically, the transcription rate of a module always starts at 0, however the total transcription rate can take on arbitrary patterns by summing multiple modules. Furthermore, the time inferred for cells in the dataset can be higher than the switch ON time of the first module, so no gene in any cell needs to have a transcription rate of 0 if this is not biologically realistic.

R1.19) Methods, Equation (24). I find the terms “biological values” and “measured values” confusing. I would rather prefer using more standard terminology, like “measured” and “expected”. Also, I think it could help readability to not use the superscripts “B” and “M”, but maybe to decorate measured values with a symbol, like a \sim or a \wedge .

Thank you for this note. We borrow this terminology from Sarkar et al., Nat Genet 2021 (DOI: [10.1038/s41588-021-00873-4](https://doi.org/10.1038/s41588-021-00873-4); ref [18] in the main text), which we feel is helpful for cell2fate. Briefly, Sarkar et al. established that observed counts in single-cell RNA seq data arise from both a biological generative process and a measurement generative process that are added together to give rise to a given mean/variance relationship. This motivates the separation of parameters in computational models into a biological (also termed “expression”) model and a measurement model. This concept has been employed elsewhere since (e.g. DOI: [10.2174/1574893618666230529145130](https://doi.org/10.2174/1574893618666230529145130) and [10.1038/s41587-021-01139-4](https://doi.org/10.1038/s41587-021-01139-4)).

We feel that this separation is also helpful in the context of velocity models. In our instance, we refer to the dynamics of mean expression as explainable by the time-evolution equations of spliced and unspliced counts as “biological” expectation values. In other words, these values would be the expected values of the Negative Binomial distribution in the absence of additional measurement noise. Effects arising from an imperfect measurement process, such as ambient RNA and varying detection efficiency across cells and genes, give rise to the “measurement” expectation values. In addition to these expectation values, the (co-) variance can in principle be separated into a biological and a measurement model as well. For simplicity, cell2fate currently does not employ a complex variance model, instead we assume that residual expression variation can be approximately described with a Gamma distribution and measurement variation with a Poisson distribution, thus giving rise to a Negative Binomial distribution. We have revised the text to improve clarity and we now define these concepts explicitly.

Implementation

R1.20) I would strongly advice against storing jupyter notebooks in the main code repository. It inflates the repo unnecessary and increases the download time. I recommend creating a separate repo for notebooks, and re-creating the code repo to remove jupyter notebooks from the history as well.

Thank you. We have refactored the code based so that all notebooks are presented in a separate repository:

https://github.com/AlexanderAivazidis/cell2fate_notebooks

They are then loaded as tutorials into in our new readthedocs.io website for example:

https://cell2fate.readthedocs.io/en/latest/notebooks/publication_figures/cell2fate_PancreasWithCC.html

R1.21) Please add unit tests and an API reference to make the package maintainable into the future, and easily useable by others.

The API reference is on the new readthedocs.io website:

<https://cell2fate.readthedocs.io/en/latest/index.html>

We have added small units tests that are run each time a commit is made to the repository. We also measure how much of the code is covered by unit tests using the codecov framework, which is similarly run each time a commit is made to the repository. The latest status of unit tests and code coverage can be directly seen on the repository home page:

<https://github.com/BayraktarLab/cell2fate/tree/main>

Installation test:

R1.22) Installation from github as described in the authors repo works. I created a new conda environment running python 3.9 and installed using ``pip install -e .``. However, when I try to import cell2fate in a jupyter notebook via ``import cell2fate``, I get

...

RuntimeError: jaxlib version 0.4.18 is newer than and incompatible with jax version 0.4.10. Please update your jax and/or jaxlib packages.

...

I tried resolving this with ``pip install --upgrade jax``, which worked.

We have now addressed this issue.

Reviewer #2:

Remarks to the Author:

The Authors present cell2fate, a novel method to perform RNA velocity analyses.

Overall, I have a very positive opinion about this work: the article is well structured and clearly presented, and the method proposed is interesting, original and well described.

The Authors convincingly validate their approach in various real datasets, comparing results to several alternative RNA velocity tools.

I think that this work represents a significant contribution to the field, and would fit well in Nature Methods.

I have several comments, but (despite the long list) all issues should be relatively easy to address (although some will take time!).

Despite the extra work, I hope the Authors can find my comments useful.

If you re-submit a revised manuscript, please highlight changes in the text so that we can easily identify them (thanks!).

Simone Tiberi, The University of Bologna

Now, the rough part.

Major comments:

R2.1) My main comment is about the presentation of the method.

While this is a methodological article, the method is only briefly and vaguely described in "The cell2fate model" Section.

From the main text, I only partially understood the assumptions, formulations and inference of the approach.

I don't think it's ok to keep all methodological aspects in "Methods".

I believe that a longer description of the method should be added in the main paper; particularly focusing on:

- **the key rationale of the approach,**

- **some key formulations such as the ODEs and their linearisation,**

- **how parameters are inferred, and**
- **the intuition behind how evidence is shared across genes, cells and modules (mentioned at the bottom of page 2).**

Is the sharing of evidence done via an Empirical Bayes approach (i.e., prior estimated from data itself)?

Thank you for these comments. It is a balancing act in which aspects of the method to include as part of the main text versus relying on a detailed description in the supplement. We have revised and expanded the methods description in the main text, taking these comments and suggestions into account, and focusing on the most important aspects and modelling choices. We would like to argue that a core innovation of cell2fate is the ability to share evidence strength across genes using velocity modules, which also greatly reduces the effective number of temporal switch parameters that need to be estimated. Please refer to page 4, for the revised methods description.

R2.2) In the Methods Section, some details are missing; in particular the full likelihood and full posterior of the model are not explicitly written.

Also, in 1.7, the “stochastic variational inference” approach, which used to infer the model parameters, is only mentioned; instead, I think its intuition of this approach and how it is applied in cell2fate should be briefly explained.

Thank you. We have added the full likelihood in section 1.5 of the supplementary methods.

We also added the full posterior in section 1.6 and a more complete explanation of our inference approach in section 1.8 of the supplementary methods.

R2.3) While Supplementary Figure 2 reports some runtimes (as far as I understand, only relative to the maximum number of modules calculation), a comprehensive computational benchmark is missing.

It would be appropriate to assess the full runtime and memory usage of the entire cell2fate pipeline (start-to-end), and of all competitors on at least 1 real dataset (possibly more).

We present extensive data on empirical (and theoretical) runtime complexity. Please see our response to comment R1.4

R2.4) I found the analysis of the “steady state system” very interesting (as in Fig 2.F).

I would appreciate a Supplementary Figure showing the results from all the other RNA velocity methods as well.

This would be useful to see if/what methods provide spurious RNA velocity directions, even in a steady state model.

Thank you for appreciating this feature of cell2fate. However we don't think a comparison of UMAP RNA velocity projections between different methods is meaningful in this case. Our reasoning is as follows.

Bergen et al. (<https://www.embopress.org/doi/epdf/10.15252/msb.202110282>) have previously identified the PBMC data as a system without any discernible RNA velocity signal.

In addition, they show in their figure 2C that scvelo's velocity graph projected onto a UMAP erroneously implies a clear trajectory in this PBMC data. We copied this figure below for your convenience. However, we do not want to make the point in our figure 2F that cell2fate's more random distribution of RNA velocity arrows is more correct, because Bergen et al. have justifiably identified this as a system without any discernible RNA velocity signal. From this perspective even cell2fate's estimates are incorrect. Instead the challenge in this dataset is rather to find a formal approach to assess the high uncertainty in RNA velocity predictions and discard the estimates based on this uncertainty. This feature would not be investigated with a comparative figure of UMAP RNA velocity projections. Instead, this is what we highlight in figure 2F, that shows even cell2fate's results should not be trusted, because our cell-specific time has a high uncertainty. We have added the reference to Bergen et al. and have revised the text to make this point more clear.

Figure reproduced from <https://www.embopress.org/doi/epdf/10.15252/msb.202110282>

Minor comments:

R2.5) At page 2, when explaining the model parameters (“the dynamics of each module ...”), all subscripts should be introduced (i.e., “cell c ”, “module m ” and “gene g ”).

We added this as suggested.

R2.6) The Authors use an ON/OFF switch model: please specify that transcription only happens in the ON state (some switch models allow for transcription to happen in both states).

The target transcription rate indeed switches from A_{mgON} to 0 after the switch off time T_{mOFF} , however the target transcription rate is approached at a rate λ_{mOFF} , so the transcription rate is non-zero for a long time even in the repression state. We hope that our rewritten method description and our reference to figure 1 makes this even more clear.

R2.7) Page 2: “as well as corresponding rates λ_{mOFF} ...” -> “as well as corresponding switch

rates λ ...”.

(Rates could refer to switch, transcription or degradation rates).

We clarified this in the text.

R2.8) All genes have the same ON/OFF switch rates; I think most (all?) RNA velocity methods share this assumption.

Nonetheless, it's still worth briefly acknowledging that this is a simplification.

We don't actually use the term “switch rates” and we wonder if our explanation of the λ rates was maybe unclear and easily confused with “switch rates” in transcriptional bursting models. Transcriptional bursting models of gene expression have switch rates, but we would like to clarify that we do not model transcriptional bursting, neither do other RNA velocity models. Instead our transcription rate can be seen as the expectation value resulting from the bursting process and our Negative Binomial noise captures any noise around this value. The λ rates govern the speed at which a module reaches its target transcription rate. Summing over all the transcription rates contributed by each module, gives the expected transcription rate for a gene over time. Hopefully our reworked method explanation in the main text clarifies this better.

R2.9) Similarly to the comment above, s and u reads are estimated and carry a degree of uncertainty.

While (as far as I am aware), this issue is not accounted for by any RNA velocity method, I think it should be mentioned.

In the conception of cell2fate, we have thought quite a bit about how to deal with uncertainty and technical factors of variation in those count estimates and so we added parameters that together make up what we called the “measurement model”. This “measurement model” contains parameters that describe the different detection sensitivity of spliced and unspliced counts as well as the noise produced by only randomly sampling them in the sequencing and quantification steps (via the Negative Binomial overdispersion parameter). This follows some suggestions in this paper: DOI: [10.1038/s41588-021-00873-4](https://doi.org/10.1038/s41588-021-00873-4) that we cite as citation [18].

R2.10) The Figures and text often refer to “counts”.

What is your definition of “counts” here: spliced only, or all counts (spliced + unspliced)?

Unless I missed it (quite possible), please clarify this explicitly both in the text, and in the software documentation/tutorial.

It's all counts, so spliced and unspliced. We have added “spliced and unspliced” at a few points in the text to make it more clear. We say “spliced and unspliced counts” both in the abstract, introduction and beginning of each section, so hopefully it's clear even if at some other points, we just say “counts”.

R2.11) You use the “cross boundary direction correctness”; I think it is worth briefly explaining what this metric is/measures.

We added a reference to the methods section at the relevant point in the main text.

R2.12) Link to the cell2fate software is missing; probably worth adding it in “Code availability”.

We have a “Code Availability” section, after the discussion and methods heading. We included more explanations in this section to clarify, where the cell2fate method itself can be installed.

R2.13) I found a few typos:

Thanks a lot, we corrected those, but did not mark them in red in the revised text for simplicity.

- Page 3 (Fig 1 caption) “allows to infer” -> “allows users to infer” or “allows inferring”
- Page 4 (title): “estimation of complex of transcription” -> “estimation of complex transcription”
- Page 7: “we expected that ... can provide ...” -> “we expect that ... can provide ...” or “we expected that ... could provide ...”
- Page 15 “A.A. conceived of the ...” -> “A.A. conceived the ...”
- Page 15 (A.A. is repeated twice) “... co.supervised A.A., A.A., ...” -> “... co.supervised A.A., ...” **That one is correct like this.**
- Methods -> please double check the punctuation after the formulas: sometimes it's missing, written on the next line, or written twice (after formula and on next line).
- Methods, formula (4) “a_g” -> “\alpha_g”
- Methods, line (186) “is denoted a module” -> “is denoted by a module”.

Suggestions (ignore if you disagree):

R2.14) If I am not mistaken, the average time the gene is active (in the ON state) is given by $\lambda_{ON} / (\lambda_{ON} + \lambda_{OFF})$.

It seems like a useful interpretation point to mention.

Actually that's not what the lambda rates are. Lambda is the rate at which a module reaches its target transcription rate. In other words, the speed of induction and repression. For clarification, these are not the switch ON/OFF rates that are found in bursting models of gene expression. The noise resulting from bursting is assumed to be adequately modelled in the negative binomial noise and our transcription rate can be seen as the expectation value of the transcription rate.

R2.15) Sub-Figures are labelled with upper case letters in the images (e.g., A), and lower case in the text (e.g., a).

I'd suggest always using the same approach for consistency.

Corrected! (But we did not mark it in red in the text, because it is such a minor change).

R2.16) In Figure 2.E (and in the software), the plot title says “Time Uncertainty”.

While “uncertainty” is an easy word to digest for biologists, I think it would be more precise if the software used what it actually plots: CV.

Ignore.

R2.17) The Git repositories with the scripts can be linked to zenodo, and associated to a DOI

Ignore.

Reviewer #3:

Remarks to the Author:

In this manuscript, Alexander et al. proposes cell2fate as a new RNA velocity model by considering time-varying unspliced mRNA transcription rates. The central assumption of cell2fate is that the gene transcription rates can be decomposed into the linear combination of gene-module specific dynamics, which are activated sequentially in various “time window” during development process. The parameter estimation was performed using Bayesian approach through pyro, and the method was compared with existing methods on real datasets. The method has also been applied to spatial transcriptomics data by inferring on non-spatial scRNA-seq counterparts first, and then project the results to spatial locations use cell2location method.

Generalizing the RNA velocity model to more complex scenarios , e.g. to incorporate cell-specific gene regulation have been developed in several existing literatures. Well-known methods include CellDancer (PMC10545816), Dynamo(PMC9332140), and VeloVAE. The **key contribution of cell2fate is to incorporate the gene module concept into RNA velocity analysis**, and use its sequential activation dynamics to reveal cell state transition processes. **This idea is novel and interesting**. However, **this feature of cell2fate seems not be fully explored in current manuscript (see comments below)**. There are also several concerns suggested to be addressed by authors before recommending its publication to Nature Methods.

We thank the reviewer for recognizing our innovations and agree that the key advance is incorporating the concept of expression modules into RNA velocity. In addition to the insights that can be generated from inspecting the modules (c.f. response to R3.4), we believe that this formulation also offers performance advantages compared to existing approaches with cell-specific transcription rates, as the modules reduce the number of parameters that need to be estimated when fitting complex transcriptional patterns across correlated genes.

R3.1) Major Concern 1: Rationale of assumption and scope of application.

The central assumption of cell2fate is that the genes are activated sequentially in modules. While the assumption seems appropriate for datasets shown in the manuscript, there are competing or alternative assumptions such as CellDancer’s relay model or the bursty model of TopicVelo (PMC10312759). It would be insightful to 1) **further compare with these** competing alternative models 2) **develop diagnostic statistics to evaluate the goodness of fit** to real dataset. In addition, the claim that cell2fate is a “biophysically accurate model” and “**mathematically sound**” seems too strong, since all RNA velocity models, including cell2fate are built upon different assumptions and their relevance to real biological scenario

might be case-dependent. Also, the word “**model-based**” in current manuscript title appears vague and confusing, given that all RNA velocity methods are somehow model-based indeed.

We have revised the presentation of the model with the aim to reduce jargon and increase the clarity with respect to the specific modelling assumptions and the implementation (c.f. Response to R2.1). We have also significantly extended our aims to benchmark cell2fate in the light of alternative methods, which now includes CellDancer (Figure 2A). We recognize there exist additional RNA velocity methods that could be considered, especially at the preprint stage. However, we feel that the set of methods we have included in the benchmark is representative of the various types of RNA velocity methods available at this point. Importantly, apart from cell2fate, none of the existing methods uses modules at the level of transcription rates. As part of the extended benchmarking, we have also extended the downstream analysis, now also using CellRank, and we compare the inferred cellular orderings of different velocity methods to known developmental stages (c.f. Response to reviewer R1, introductory paragraph on revised benchmarking strategy).

Regarding a conceptual comparison of cell2fate with alternative RNA velocity methods, we agree with the reviewer that cell2fate stands out in the way the velocity problem is factorised into modules. We would like to draw the referees attention to section 2 of the supplementary methods, which includes both an in-depth discussion and a short summary in table (figure 3) to expose the similarities and differences between established RNA velocity models. It is certainly correct that cell2fate does also employ biophysical approximations. For example, the model does currently not account for stochastic bifurcations and assumes constant degradation rates. Having said this, we feel it is fair to say that previous approaches are subject to even stronger biophysical simplifications. Cell2fate solves the challenge of capturing gene and cell-specific transcription rates with a closed-form solution for the time evolution of spliced and unspliced counts with module-level parameters that are shared across genes and thus improve both interpretability and statistical power. This design choice also renders the model easily extendable. Nonetheless, we have adapted the language in the manuscript and toned down the corresponding claims, for example replacing absolute statements such as “biophysically accurate” with the relative statement “more biophysically accurate”.

Concerning point (2) we agree that the ability to diagnose the model will be important to render the methods practically useful. We have extended the main text and added a second tutorial notebook to the documentation, specifically focussed on assessing the confidence in cell2fate’s predictions, available here:

https://cell2fate.readthedocs.io/en/latest/notebooks/publication_figures/cell2fate_AssessingConfidence.html

We expect that simple goodness of fit statistics, such as the total mean squared error, may not be sufficient for more complex RNA velocity models, since they can provide a good fit purely due to their large number of parameters compared to simpler models. Instead, we propose the posterior of cell2fate’s cell specific time to assess whether cell2fate’s parameters are identifiable in a given dataset, which is not the case in the PBMC dataset for example (Figure 2F). We have now further extended this analysis. Firstly, we provide a

function to plot the posterior time estimates of two clusters involved in putative cell state transitions. To illustrate this general approach, we present an example from the tutorial notebooks below in figure R3.1. The example shows that the transitions from clusters HSC_1 to Ery_1 and Ery_1 to Ery_2 are based on time estimates with distributions that overlap only sparsely. In contrast, the HSC_1 to HSC_2 transition - which is the only transition cell2fate fails to estimate correctly - is based on strongly overlapping distributions. Secondly, we summarised this approach into a confidence metric that indicates what percentage of posterior samples in the second cluster have higher time values than the 90th percentile of cells in the first cluster. We have included this metric in table R1 for all transitions (supplementary table 8 and Supp. Fig. 23) and discuss it in section 2 of the paper.

Figure R3.1: Posterior samples of time estimates of cells in clusters involved in cell state transitions in the Human Bone Marrow dataset. Red indicates the cells assumed to be in the target cluster based on prior knowledge.

In addition, the absolute values of gene-specific splicing/degradation rates (beta and gamma) of each gene is not possible to solely determined from scRNA-seq data (PMC9332140)– this is a well-known problem about “invariant of scales” in RNA velocity models (pubmed: PMC9550177). Would this bring challenge for the foundation of cell2fate? Could author explain more that how cell2fate overcomes such limitation and obtain the gene-shared unit time?

We agree that this is an important limitation, which however applies to all current RNA velocity methods. An implication of this limitation for our gene-shared unit time is that it cannot estimate the absolute value of temporal intervals in the data. However, we expect that our time measure could provide a relative comparison between two time intervals. For example, based on our time estimates in the Dentate Gyrus dataset, the Radial-glia like to Astrocyte trajectory is shorter than the trajectory from nIPCs to Granule mature cells, which agrees well with prior knowledge of Astrocytes being born later in development and thus having less time to mature by the time point sampled in the dataset.

One approach to obtain more accurate estimates of absolute time intervals is to set the relevant priors to realistic values. We use this approach for the priors of the splicing and degradation rates that are in the same order of magnitude as prior knowledge, specifically around 1 molecule per hour. As a consequence we expect the gene-shared unit time to converge to the correct order of magnitude. In our response to R1.1) we also found that our gene-shared time correlates well with the developmental time of samples in the Dentate Gyrus and Erythroid Maturation datasets. An implication of this result for future

developments is that an informative prior could be set on the gene-shared unit time, based on the developmental age of samples, but we did not explore this in the current manuscript.

R3.2) Major Concern 2: Systematic benchmarking with ground-truth.

The current manuscript has compared the algorithm with existing methods only on real datasets, using CDir metric. This metric seems indirect given that the ground truth velocities in the datasets are not known quantitatively in real dataset. Could the author include more quantitative benchmarking on simulated datasets where ground truth velocities? In addition, what about the scalability of cell2fate with the increasing size of dataset?

We agree with the reviewer that a robust benchmark of cell2fate is required. As the generation of simulated datasets is challenging, at least without biasing the outcome to the data generation process, we primarily focus on real data benchmarks using different metrics. In response to comments R1.1 and R1.2, we have substantially revised and extended this assessment, which now includes a comparison to the developmental time point of sampling and known cell state transitions, as well as a downstream analysis using CellRank. (c.f. Response to reviewer R1, introductory paragraph on revised benchmarking strategy). These additional results are displayed in Supp. Figures 14-17, 20 and 21, and largely confirm the previous conclusions

Regarding simulations, we have conducted a focused simulation study with the aim to assess the extent to which different properties of the (simulated) generative processes affect the accuracy of the estimates obtained from cell2fate. Briefly, we first fit cell2fate to the Dentate Gyrus dataset and then generated semi-synthetic datasets from the cell2fate generative model keeping all parameters at their fitted values except for one parameter which we varied (by means of multiplication with 0.25, 0.5, 1, 2, or 4). We then assessed the correlation between the true simulated splicing and degradation rate and the estimated values (since RNA velocity is a function of those rates).

We chose to investigate four key biological and technical parameters in this way: the splicing rate, degradation rate, detection probability of transcripts and Negative Binomial overdispersion parameter (Figure R3.2). We observed robust results across these different settings. Lower correlations were obtained, when the splicing rate became much larger than the degradation rate, when the noise in the data was increased (low overdispersion parameter) and when the detection efficiency of transcripts was lowered. We added this

analysis as supplementary figure 19.

Figure R3.2: Correlation of cell2fate inferred splicing and degradation rates compared to ground truth simulated from the cell2fate generative model with different parameter choices. Results on simulated data are shown with A: varying splicing rates across genes B varying degradation rates across genes C varying overdispersion parameter (noise) across genes D varying detection efficiency across cells

Regarding the computational scalability of cell2fate, please refer our response to R1.4, where additional theoretical and empirical arguments are presented, supporting that cell2fate is sufficiently efficient for most use cases that occur in practice (up to ~1 million cells).

R3.3) Major Concern 3: Sensitivity analysis and robustness test.

In the proposed algorithm, there are numerous heuristic choices of parameters especially regarding the prior distribution. Another important hyper-parameter is the max number of modules and is selected empirically by leiden cluster number times 1.15. However, their influence on performance and algorithm robustness has not been studied in current manuscript. Extensive study on parameter sensitivity and cell2fate robustness seems very necessary for the broader application of the method.

We agree that robustness is important. We have slightly refined our experiments, so this revision now means that all experiments and results presented throughout the manuscripts are conducted without the need for tuning of (hyper) parameters. Specifically, we previously used a larger expected mean and standard deviation (500 and 150) for the maximal time parameter (Tmax) for the Human Bone Marrow dataset, but now we use the same value as in the other 7 datasets shown in our study (50 and 50). Thus, there is now no need to tweak the model to different dataset or settings. The sole parameter that needs attention is the number of modules that is changed for each dataset, which however can be determined using a fully automated heuristics. Having said this, it is of course interesting to understand the impact of key hyperparameters we propose. To address this, we have carried out a robustness analysis by iterating over key prior distribution parameters in our model, as well as the number of chosen modules and highly variable genes for the Dentate Gyrus dataset and computing the CBDIr metric each time. Briefly, for each parameter we chose two values above and two values below our default values, where the default values are indicated with a

star. See our new supplementary figure 18 below, where we show both the average CDir metric and the performance on individual transitions in the dataset.

Reassuringly, we note that the number of modules and genes has remarkably little effect on the performance, as long as these parameters are not too low, i.e. less than one module per cluster or less than 2000 highly variable genes. We find that it increases performance to set the prior distribution on the maximal time in the dataset to a lower mean, but a higher standard deviation. Concerning dynamical rates, we found that expecting splicing and degradation rates to be the same (at 1 molecule per hour) or a splicing rate slightly lower than the degradation rate gives the best results. We find only small effects of varying the expected noise and detection efficiency in the data. Overall, we see that transitions involving highly abundant cell types, such as Neuroblasts and Granule neurons are less susceptible to suboptimal prior distribution settings (blue dots and lines) than rare cell types, such as OPCs (yellow dots and lines).

The results also indicate that further optimization to determine parameters in a data-specific manner could be considered, for example decreasing the prior expected mean of the T_{max} parameter and increasing its variance. Such “tuning” always comes at the cost of increased complexity of the model and the risk of overfitting. Thus, at this point we recommend running the model with default values.

Figure R3.3: Performance (CDBir values) of cell2fate on the Dentate Gyrus dataset with various preprocessing choices and prior distribution settings. The x-axis label shows the name of the prior distribution that was varied in the first row and the symbol of the parameter for which the prior distribution was varied in the second row. In addition, the number of

modules to the number of Leiden clusters factor was varied and the number of input genes (row 2, left and middle).

R3.4) Major Concern 4: New biological insights.

Considering gene module dynamics as a key novelty of cell2fate, the downstream analysis on its biological implication and insights seems not adequate in current manuscript. For instance, could module dynamics imply about the “driver genes” in the transition? Are there specific genes that have significant components in multiple modules, and what is the biological interpretation? In other words, if the gene expression dynamics are all solely dominant by one specific module, **would the cell2fate simply reduces to scVelo model, by assuming a single on-off switch?** Explorations like this could greatly enhance cell2fate’s interests to general readers and validate its rationale of sequential module activation assumption.

We agree that the factorization of the velocity problems using modules is a key innovation, and in fact we have exploited them throughout the paper. Briefly, in figure 2D we show that the modular transcription rate allows estimating “transcriptional boosts” in erythroid maturation, i.e. multiple changes in the transcription rate over time. These “multi-rate kinetic genes” would correspond to the genes you mention with “significant components in multiple modules”. Moreover, the comparison to the pyroVelocity_model2 in figure 2C and D illustrates that a simpler stepwise model (as also done in scvelo) is insufficient in this case to capture the complexity of the dynamics. Next, we have exploited the modules in figures 3B-H to illustrate how they can decompose a trajectory into interpretable components and suggest “driver genes” in the form of module marker genes and transcription factors. We would also like to highlight at this point an analysis that we have already done, but only included as supplementary figure 29. In particular, we have investigated whether the top 20 module transcription factors in the granule neuron differentiation trajectory have their binding targets in the promoter regions of the module marker genes. For this purpose, putative promoter sequences were first extracted in the genomic vicinity of the top 300 module genes of each module. Binding affinities were then predicted between these sequences and top 20 module transcription factors using the ProBound algorithm. Finally, we have exploited the modules in figure 4 to integrate single-cell transcriptional dynamics with spatial transcriptomics.

Cell2fate has a convenient function to automatically extract top module genes and transcription factors, termed “get_module_top_features”:

https://cell2fate.readthedocs.io/en/latest/cell2fate.html#cell2fate.Cell2fate_DynamicalModel.get_module_top_features

We also provide an example of it’s usage in the Dentate Gyrus tutorial notebook:

https://cell2fate.readthedocs.io/en/latest/notebooks/publication_figures/cell2fate_DentateGyrus.html#:~:text=This%20method%20returns,for%20all%20modules%3A

In addition to marker genes and TFs of each module, the function also provides gene set enrichment results. We have now added the output of this analysis as supplementary table 9.

Finally, in response to your comment we have now added an additional exploratory analysis, intersecting the top 30 module genes with risk genes for Alzheimer's and autism. Interestingly, this analysis highlighted the late granule neuron maturation module 8, where both Alzheimer's associated gene *Trappc6a* and Autism associated genes *Kdm6a* and *Nr4a2* are among the top 30 markers.

R3.5) Major Concern 5: Detailed documentation and API.

The provided github repo includes the annotated jupyter notebook for tutorial along with codes to reproduce results. In the meantime, the major improvements are needed to include detailed documentation about functions and APIs in cell2fate package.

We improved this aspect of the method and provide a read-the-docs website with detailed documentation about functions and APIs:

<https://cell2fate.readthedocs.io/en/latest/>

R3.6) Major Concern 6: The Methods file needs significant revision to meet Nature Methods standard

1) The Methods file is not well formatted, along with quite some typos. Overall, it appears overly technical and lengthy for general readers of Nature Methods – the authors may consider split some part into supplementary notes. Format-wise, many sentences involving equations are starting with the comma (,), e.g. line 396. There's also issue with paired double quotation marks (""), e.g. line 382. Other typos include the notation error on line 465 and reference error line 470.

2) Some claims seem highly speculative to appear in Methods. For example, the comment about stochastic modelling (line 250-253) lacks convincing validation.

We have significantly revised the presentation of the model and split the methods file into two parts. A brief description of the model is presented as "Methods" as part of the main text. This section presents a concise description of the cell2fate model, as well as the data analysis and experimental methods description. A full and detailed methods description is presented as Supplementary Methods, which presents our previous and significantly revised in depth model description and comparative analysis to other RNA velocity methods. We have also toned down any speculative claims on future model extensions, replacing these with more concrete near-term goals.

R3.7) Minor :

1. Relevant to Major point 2, does the proposed method filter "velocity genes" based on "goodness of fit" (similar to scVelo) to plot streamlines? Does different highly-variable gene selection affects the results?

We do not filter “velocity genes” for any downstream analysis. We included different numbers of highly-variable genes in the robustness analysis in supplementary figure 18 and we do not find it to affect results strongly if the number is equal to or above 2000.

2. The gene names in Supp. Fig. 23 are too small to be recognized.

We fixed this with a higher resolution figure.

3. Have the gene expression values in Fig. 3B and Fig. been smoothed?

The gene expression values represent the total counts produced by each module as estimated by the cell2fate model, so they are “smoothed” in the sense that they are a model estimate rather than raw data.

Reviewer #1 (Remarks to the Author):

The reviewers greatly improved their manuscript and satisfied almost all of my previous comments. The only (small) point left from my side is their scalability evaluation (both for time and memory) - I think this can be visualized in a much more effective way than in the current tables, which make it hard to interpret scalability in the number of cells and genes (consider e.g. scatter plots, bar charts - anything that makes it possible to evaluate scalability visually). The authors claim linear scalability, it would be nice to also demonstrate this practically (the authors have the results already).

Also, methods that make use of parallelization (like cellDancer) should be run in parallel - it seems unfair to run GPU-optimized methods on GPU, but to not run CPU-optimized methods in parallel, if they support parallelization.

In summary, I'm happy with the revised version and I recommend publication in Nature Methods.

Reviewer #1 (Remarks on code availability):

I've reviewed the code in the first iteration and the authors have resolved my comments.

Thank you for your comments. We have tried to rerun the cpu based methods with more cores, but we ran into problems with the first method we tried, CellDancer and at the point of writing our issue is not yet resolved:

<https://github.com/GuangyuWangLab2021/cellDancer/issues/33>

In addition, others have reported problems when running the method in parallel:

<https://github.com/GuangyuWangLab2021/cellDancer/issues/7>

Furthermore, a question that remains would be how many cores to use. Using 64 cores would be significantly faster than just 8 for example.

Hence, as an alternative, we have now plotted execution times for gpu and cpu based methods separately in supplementary figure 23 A and B and we have added a clear note to figure 23B that says that only 1 job was run and that the execution time for cpu based methods can in principle be expected to decrease linearly with the number of jobs run.

We also made a sequence of cell2fate runs on the Erythroid maturation datasets using variable numbers of modules, genes and cells to show approximately linear scalability of the execution time empirically, although some noise around this linear relationship is clearly present as well. These results are included in supplementary figure 23 C, D, E. For convenience we include supplementary figure 23 below:

A**B****C****D****E**
Supp. Figure 23: Execution time of methods on different datasets **A:** Execution times of gpu-based methods on all datasets **B:** Execution times of cpu-based methods on all datasets using 1 job. Execution times can be expected to decrease linearly with the number of jobs used. **C:** Execution time for cell2fate as a function of cell numbers randomly sampled from the Erythroid maturation dataset fixed at 3000 genes and 6 modules. **D:** Execution time for cell2fate as a function of highly variable genes using the Erythroid maturation dataset fixed at 3000 randomly sampled cells and 6 modules. **E:** Execution time for cell2fate as a function of number of modules using the Erythroid maturation dataset fixed at 3000 genes and 3000 cells.

Reviewer #2 (Remarks to the Author):

The Authors have suitably implemented (or replied to) all my comments.

I'd like to thank the Authors for the time and effort they invested to implement the suggested corrections and clarifications.

In the past, I've worked with ON/OFF 2-state transcription models, and I had partially misunderstood some aspects of cell2fate (interpreting it based on my previous knowledge). The added explanations make the context clearer.

I only have an additional minor request regarding the mathematical rigour of the formulations in the Supplementary (and partially in the main text).

- distributions are usually expressed with tilda (\sim) instead of equals ($=$).
E.g., $X \sim \text{NegativeBinomial}(\dots)$ (formula (8)).

- densities are usually denoted by $f(\dots)$.
e.g. (at Supp formula (37)) $P(X = x; \theta) = f_{\text{NB}}(x; \theta)$, where $f_{\text{NB}}(\cdot; \theta)$ indicates the density function of a Neg Bin distribution with mean ... and variance

- the likelihood in Supp formula (37) is only written with respect to X_{cgj} .
That is the density of X_{cgj} though, not of the entire dataset.
The full likelihood is likely expressed as the (possibly treble) product of that density over indices c , g and j .

Furthermore, θ lacks subscripts; since θ varies depending on subscripts cgj , I think it'd be written as θ_{cgj} for X_{cgj} .

While the full θ vector is likely the aggregation of such θ_{cgj} elements (please define it clearly).

Typo:

- Page 4: which vastly reducing -> which vastly reduces

P.s., "Ignore" is the more fun reply I ever received as a reviewer.

Fair enough: after replying to 15 comments, it seems comprehensible.

Nonetheless, I still think that in the caption of Figure 2 the word "uncertainty" should be clarified (uncertainty is a vague term that can be measured in multiple ways).

Simone Tiberi,
The University of Bologna

Thank you for your comments. First of all, please accept our apologies for the "Ignore" reply to two of your previous minor comments. Since you prefaced the suggestion with " (ignore if you disagree)" we put down "Ignore" as a mental note and planned - but neglected - to expand on our reply later. We preferred to keep the heading inside the figure 2F simple and abstract and then explain in more detail in the figure legend 2F that it is the coefficient of variation. However, we have now also added "(CV)" inside the figure. We also added cell2fate to Zenodo here: <https://zenodo.org/records/13883214>

Thank you for your suggestions on mathematical notations and the typo. We have changed the relevant parts in the main text and supplementary methods accordingly.

Reviewer #3 (Remarks to the Author):

In the revised manuscript, the authors have done a good job of addressing the previous concerns, especially on improving the robustness of method. I have no further questions and recommend the acceptance of this manuscript.

Reviewer #3 (Remarks on code availability):

The cell2fate package is a usable resource for the community with enough instructions for installing and running the applications. I have no problem to run the test codes on GPU.